# v1: Learning to Point Visual Tokens for Multimodal Grounded Reasoning

## Abstract

When thinking with images, humans rarely rely on a single glance: they revisit visual information repeatedly during reasoning. However, existing models typically process images only once and thereafter generate reasoning entirely in text, lacking mechanisms to re-access or ground inference in visual representations. We empirically confirm this: as reasoning chains lengthen, models progressively lose focus on relevant regions. In response, we introduce **v1**, a lightweight extension that enables active visual referencing through a simple point-and-copy approach. This allows the model to identify relevant image patches and copy their embeddings back into the reasoning stream, ensuring that evolving hypotheses remain grounded in perceptual evidence. Crucially, our pointing strategy lets the MLLM directly select image patches using their semantic representations as keys, keeping perceptual evidence embedded in the same space as the model's reasoning. To train this capability, we construct **v1g**, a dataset of 300K multimodal reasoning traces with interleaved visual grounding annotations. Across various multimodal mathematical reasoning benchmarks, **v1** consistently outperforms comparable baselines, establishing point-and-copy as a practical mechanism for grounded reasoning. We will release the model checkpoint and data.

## 1 Introduction

When reasoning with images, people rarely rely on a single glance. A student solving a geometry problem may repeatedly consult the diagram, checking angles, points of tangency, or symmetries while refining their inferences. Findings from cognitive science support this intuition: humans often revisit visual information to uncover new details, adjust interpretations, or externalize reasoning through sketching (Cox, 1999; Brun et al., 2016; Chu et al., 2017; Kozhevnikov et al., 2002).

Recent progress in Multimodal Large Language Models (MLLMs) (Liu et al., 2023a; Bai et al., 2025; Chen et al., 2025) has extended language models with the ability to process images alongside text. Further, MLLMs are finetuned for multimodal reasoning (Xu et al., 2025; Yao et al., 2024; Sun et al., 2025; Huang et al., 2025), where models must integrate visual and textual information through multi-step inference rather than direct recognition or description. A primary example is multimodal mathematics (Lu et al., 2024; Zhang et al., 2024a; Wang et al., 2024a), which requires explicit multi-step reasoning over visual and textual information and provides unambiguous solutions.

However, current MLLMs process images only once at the start and, due to causal masking, thereafter reason mainly over the frozen key–value cache of visual embeddings. This limits their ability to actively revisit visual context as inference unfolds. In practice, this constraint manifests as two forms of *visual grounding decay*. First, attention to all image tokens steadily weakens as reasoning chains extend. Second, even the relative weight on relevant tokens declines, reducing the model's ability to focus on the most informative regions (section 3). These effects highlight the need for mechanisms that let models actively re-access visual information to keep reasoning grounded in the input.

To this end, we propose **v1**, a simple yet effective extension that equips MLLMs with a *point-and-copy* mechanism for dynamically referencing input visual tokens during multimodal reasoning (fig. 1). Specifically, we augment the model with an additional pointing head that outputs a probability distribution over the input image token positions, alongside the standard vocabulary logits. When an image token is selected, its embedding is copied and injected as the next-step input embedding, enabling the model to dynamically retrieve and reuse visual information during generation.

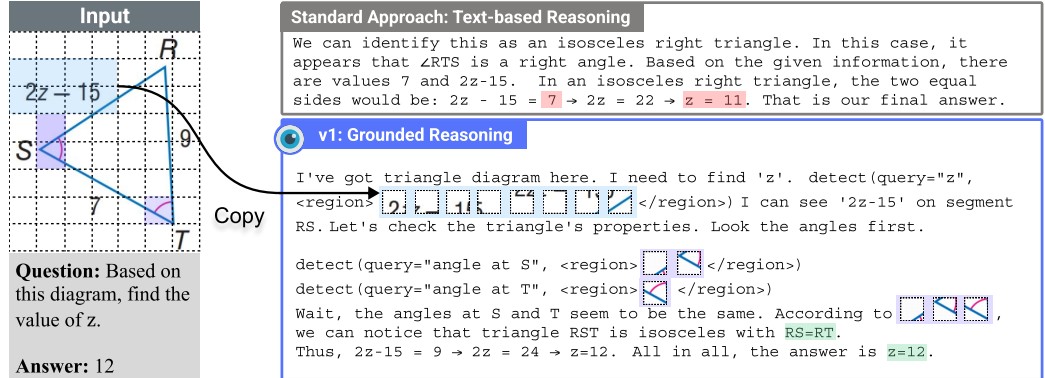

Figure 1: **Pure text-based reasoning *vs*. v1 during inference.** Our **v1** can actively re-access visual context by pointing to and copying relevant image regions throughout the reasoning process.

Our approach is readily compatible with popular MLLM architectures (Liu et al., 2023a; Wang et al., 2024c; Chen et al., 2025) that operate on continuous image embeddings. Unlike methods that attempt to generate new image tokens (Team, 2024), which are often computationally intensive and prone to instability, our method simply reuses existing input embeddings through pointing and copying. The only additional parameters are lightweight linear heads, incurring minimal computational overhead.

To train **v1**, we construct **v1g**, a dataset of 300K multimodal reasoning paths with interleaved grounding annotations, where each reasoning step is explicitly linked to a corresponding image region. The construction pipeline comprises three stages: (1) oversampling diverse reasoning traces from an MLLM, (2) extracting visual queries and retrieval steps from the traces using an LLM-guided decomposition process, and (3) grounding each visual reference by associating it with a bounding box in the input image. The pipeline is fully automated, leveraging the generative and interpretive capabilities of LLMs to produce high-quality, grounded reasoning trajectories at scale.

We evaluate **v1** on three established multimodal mathematical reasoning benchmarks: MathVista (Lu et al., 2024), MathVision (Wang et al., 2024b), and MathVerse (Zhang et al., 2024a), following prior work (Yao et al., 2024; Sun et al., 2025; Huang et al., 2025). **v1** demonstrates strong performance across all benchmarks, outperforming existing models of comparable scale and approaching the capabilities of much larger models, particularly on tasks requiring precise visual grounding and iterative reference to localized regions. These results suggest that dynamic access of visual input at inference time can improve multimodal reasoning capabilities.

Our contributions are:

- **v1 model:** a lightweight MLLM extension that mitigates visual grounding decay through a novel point-and-copy mechanism, enabling dynamic visual reference.
- **v1g dataset:**, a large-scale training set with 300K multimodal reasoning traces and fine-grained visual grounding.
- **Empirical findings:** extensive experiments and ablations on multimodal mathematical reasoning benchmarks, showing that dynamic visual reference and the point-and-copy design both mitigate visual grounding decay and lead to better multimodal reasoning.

## 2 RELATED WORK

### 2.1 REASONING IN LARGE LANGUAGE MODELS

**Reasoning in text-only large language models.** The introduction of OpenAI's o1 model (Jaech et al., 2024) marked a breakthrough in LLM reasoning, achieving unprecedented performance on mathematical benchmarks (Lightman et al., 2023; Cobbe et al., 2021). This success sparked widespread efforts to reproduce and enhance these capabilities, with DeepSeek-R1 (Guo et al., 2025) demonstrating how reinforcement learning can promote reflective Chain-of-Thought behaviors, and subsequent work exploring inference-time scaling to encourage deeper reasoning (Muennighoff et al.,

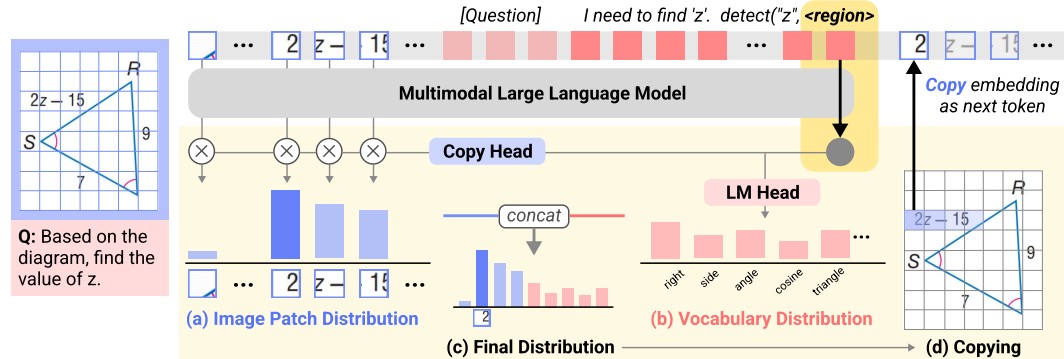

Figure 2: **Inference process of v1.** At each step, the MLLM encodes the multimodal context and generation history into token representations. For the last token (e.g., "<region>"), (a) a copy head projects its representation and computes logits against image patch embeddings, (b) a language head produces logits over the vocabulary, and (c) the two are concatenated to form the final distribution. If a patch is chosen, its embedding is copied as the next token input, enabling **v1** to reference image regions one patch at a time.

2025). While these advances have transformed text-based reasoning, extending such capabilities to multimodal settings introduces new challenges.

**Reasoning in multimodal large language models.** Multimodal reasoning poses challenges beyond text-only inference, demanding both raw perception and the integration of visual inputs into reasoning. Prior approaches (Liu et al., 2023a; Chen et al., 2024; Zhang et al., 2024b; Yang et al., 2023) often convert visual content into descriptive text for downstream reasoning. More recently, inspired by Chain-of-Thought prompting in LLMs, models such as LLaVA-CoT (Xu et al., 2025), Mulberry (Yao et al., 2024), Vision-R1 (Huang et al., 2025), TVC (Sun et al., 2025), OpenVLThinker (Deng et al., 2025), and MM-Eureka (Meng et al., 2025), among others (Zhang et al., 2024b; Wu et al., 2025; Wang et al., 2025), extend CoT reasoning to multimodal tasks and achieve strong results on benchmarks such as MathVista (Lu et al., 2024) and MathVision (Wang et al., 2024b). However, these models treat the image context as fixed input and then carry out reasoning entirely in the text space, without an explicit mechanism to re-access or ground their reasoning in visual representations.

## 2.2 IMPLEMENTING VISUAL REFERENCE

Humans reason with images by actively engaging with specific regions, often revisiting or sketching them to support problem-solving (Cox, 1999; Brun et al., 2016; Chu et al., 2017; Kozhevnikov et al., 2002). Likewise, MLLMs should support step-wise interaction with visual inputs; either by referencing regions (Gupta & Kembhavi, 2023; Hu et al., 2024) or generating intermediate visuals (Borazjanizadeh et al., 2025). We briefly review prior approaches to this challenge.

**Coordinates.** Some MLLMs are trained to output bounding boxes to refer to relevant image regions (Gupta & Kembhavi, 2023; Wu & Xie, 2023). While effective in constrained settings, this approach resembles a "call-by-key" mechanism—accessing visual content via position. It depends on accurate detection and fails in cases where relevant visual cues are abstract or not spatially localized.

**Image generation.** Other methods (Li et al., 2025; Borazjanizadeh et al., 2025; Ma et al., 2025) allow models to externalize reasoning by rendering intermediate visual states or generating new images. While expressive, these approaches are limited to programmatic rendering or require full generative pipelines, which add significant computational overhead. Furthermore, bridging discrete image tokens (*e.g.* VQ-VAE (Van Den Oord et al., 2017)) with continuous vision-language embeddings introduces representational mismatch, complicating integration.

**Pointing.** We build on the Pointer-Generator Network (PGN) (See et al., 2017), originally developed for selective text copying, and extend it to the multimodal setting. Our method allows MLLMs to

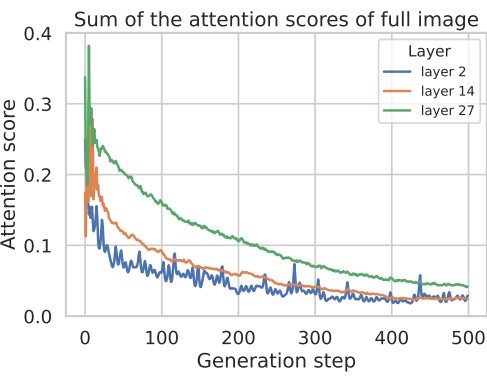 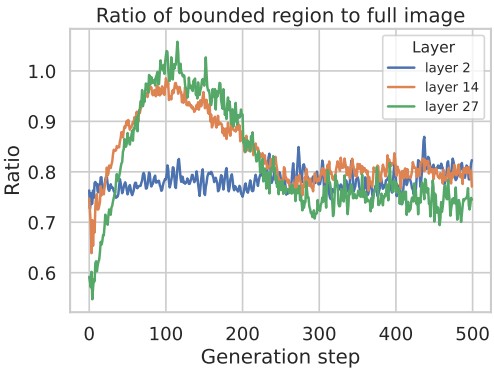

(a) Cumulative attention across all visual tokens. (b) Attention ratio: salient regions *vs.* all visual tokens.

Figure 3: **Attention dynamics during reasoning.** (a) illustrates a gradual decrease in overall attention to the input image tokens, while (b) indicates that semantically important visual regions receive disproportionately low attention, suggesting inefficient grounding during reasoning.

dynamically point to and reuse image embeddings during generation, enabling direct and interpretable visual re-access without coordinate prediction or image synthesis.

## 3 VISUAL GROUNDING DECAYS DURING REASONING

To examine how visual attention evolves at each generation step, we use RefCOCO (Kazemzadeh et al., 2014), a visual grounding benchmark in which each example consists of an image and a target region defined by a bounding box. The task requires generating a caption that uniquely identifies this region, offering a natural probe for whether a model attends to the correct visual context. Though RefCOCO generally involves shorter generations than mathematical reasoning, it provides a controlled setting to measure attention dynamics with ground-truth visual regions. We analyze the TVC-7B model (Sun et al., 2025) on the RefCOCO testA split, focusing on attention weights between the most recently generated token and all image tokens. Layers 2, 14, and 27 of the 28-layer transformer are analyzed to represent early, middle, and late stages of MLLM processing.

The first analysis (fig. 3a) tracks the total attention allocated to image tokens at each decoding step. In all layers, attention steadily declines, suggesting a shift from visual grounding to reliance on internal memory as generation proceeds. The second analysis (fig. 3b) measures attention to the task-relevant region by computing the ratio of mean attention within the bounding box to mean attention across all image tokens. Layers 14 and 27 initially show increased focus, but by mid-generation, all layers converge to a ratio of ~0.8, indicating that salient tokens receive less attention on average than background tokens.

These findings suggest *visual grounding decay*, where models progressively lose attention to visual content during extended generation. This limitation is of particular relevance to multimodal reasoning domains such as mathematics, where extended reasoning chains necessitate repeated and precise reference to diagrams. These observations motivate architectures that incorporate dynamic visual access during inference to maintain grounding and enhance multi-step reasoning.

## 4 METHOD

### 4.1 PRELIMINARY: POINTING FOR LANGUAGE GENERATION

**Formulation.** We formulate a conditional next-token prediction objective, as commonly adopted in modern multimodal large language models (MLLMs). Given a sequence of continuous input representations $c$ (*e.g.* embedded text tokens or visual features) the model is trained to autoregressively

predict the discrete next token $x_t$ conditioned on the input $c$ and previously generated tokens $x_{<t}$:

$$p(x_1, \ldots, x_T \mid c) = \prod_{t=1}^{T} p(x_t \mid c, x_1, \ldots, x_{t-1})$$

The continuous input sequence $c$ may include a heterogeneous mixture of modality-specific features, such as embedded discrete text tokens or continuous visual embeddings produced by image encoders (*e.g.* CLIP (Radford et al., 2021)). This general formulation covers a wide range of multimodal architectures such as LLaVA (Liu et al., 2023a) and Qwen-VL (Bai et al., 2025), which use continuous input image representations.

**Pointing.** For visually grounded reasoning, it is often necessary to refer back to a specific region or token within the input sequence $c$, especially when that region corresponds to localized visual content. Rather than generating a new description of a visual entity, we may instead wish to *point* to its position within the input, thereby referencing it explicitly.

The pointing mechanism we examine was first introduced by the pointer-generator network (See et al., 2017) in text summarization research. In the pointer-generator network, the input context sequence $c$ also consists of discrete tokens within the vocabulary space $V$, unlike our setup. The model dynamically mixes two distributions at each decoding step $t$: (1) a generation distribution over the vocabulary $P_{\text{gen}}(x_t)$, and (2) a copy distribution $P_{\text{ptr}}(x_t)$ over input tokens. The final output probability is given by a gated mixture:

$$p(x_t \mid c, x_{<t}) = \lambda(x_t \mid c, x_{<t}) \cdot P_{\text{gen}}(x_t \mid c, x_{<t}) + (1 - \lambda(x_t \mid c, x_{<t})) \cdot P_{\text{ptr}}(x_t \mid c, x_{<t})$$

where $\lambda \in [0, 1]$ is a learnable scalar gate that controls the trade-off between generating a new token and copying one from the input.

The pointer distribution is obtained via attention over the encoder representations:

$$\alpha_t^{(k)} = \frac{\exp(\text{score}(h_t, c_k))}{\sum_{k'} \exp(\text{score}(h_t, c_{k'}))}, \qquad P_{\text{ptr}}(x_t = w) = \sum_{k : w_k = w} \alpha_t^{(k)}$$

where $h_t$ is the decoder hidden state at step $t$, $w_k$ the token at position $k$, and score denotes a standard attention scoring function (*e.g.*, dot-product or additive). We generalized the formulation beyond the original implementation to arbitrary autoregressive language models for explanation purposes.

**Discrete targets.** The above formulation constrains the pointing targets to be within the discrete vocabulary space $V$. This prevents application to general MLLMs as the multimodal inputs often consist of continuous feature sequences (Liu et al., 2023a; Bai et al., 2025).

### 4.2 V1: POINTING FOR MULTIMODAL GROUNDED REASONING

To overcome these limitations, we introduce **v1**, a lightweight extension to autoregressive MLLMs that enables explicit grounding by pointing to continuous input representations. **v1** augments the standard vocabulary with pointer tokens that reference input embeddings, allowing the model to either generate text or copy visual content on demand. All functionalities, including textual reasoning and visual grounding, are integrated into a single finetuned backbone model (e.g., Qwen-2.5-VL) without relying on any external module or auxiliary grounding network during inference. As a result, **v1** supports inference over both discrete and continuous modalities in a unified framework without requiring modifications to the model's core architecture. Figure 2 illustrates its inference process.

**Pointing to continuous inputs.** The gated mixture formulation of See et al. (2017) is not directly applicable to continuous inputs as image embeddings, as such inputs lack discrete mappings to vocabulary tokens $V$. To enable pointing in this setting, we extend the output space to include references to positions in the continuous input. Specifically, we define the augmented output space as $\bar{V} = V \cup C$, where $C = \{c_1, c_2, \ldots, c_K\}$ denotes the set of continuous input vectors (e.g. embeddings of the input image patches). This formulation allows the model to generate either a vocabulary token or a pointer to a specific continuous input. We denote a pointer to input vector $c_k$ as $\langle \text{ptr} : c_k \rangle$, which is treated as a discrete token during decoding.

At each decoding step $t$, the model computes two distributions: (1) a generation distribution over the vocabulary $V$, producing logits $\text{logit}_{\text{gen}} \in \mathbb{R}^{|V|}$, and (2) a pointing distribution over the input positions $C$, producing logits $\text{logit}_{\text{ptr}} \in \mathbb{R}^K$. The final output logits are defined as:

$$\text{logit}_t = \left[ \text{logit}_{\text{gen}} \,\|\, \text{logit}_{\text{ptr}} \right] \in \mathbb{R}^{|V|+K}$$

where $[\cdot \,\|\, \cdot]$ denotes concatenation. Pointing logits are computed by attending over the input sequence:

$$\text{logit}_{\text{ptr}}^{(k)} = \frac{L_q(h_t) \cdot L_k(c_k)^\top}{\sqrt{D}}$$

where $h_t$ is the decoder hidden state at step $t$, $L_q$ and $L_k$ are learned linear projections, and the scaling factor $\sqrt{D}$ follows standard attention practice. We omit the gating module $\lambda$ used in previous work, as the logit types are defined over disjoint spaces and do not require interpolation.

During inference, if the model selects an index in $V$, the next token $x_t$ is emitted as the corresponding vocabulary token. If the model selects an index $k \in C$, the token is represented as a pointer $x_t = \langle \text{ptr} : c_k \rangle$. On the subsequent decoding step, the input embedding at position $t$ is replaced with the continuous vector $c_k$, enabling the model to attend directly to the referenced content.

### 4.3 ANNOTATING VISUALLY-GROUNDED REASONING DATA

To train **v1**, we require fine-grained multimodal reasoning traces in which each step is grounded to specific visual evidence. To this end, we construct **v1g**, a dataset of 300K multimodal reasoning paths with interleaved grounding annotations. Each trajectory includes a sequence of reasoning steps, where textual inferences are explicitly linked to corresponding image regions. The dataset is generated through a fully automated three-stage pipeline: (1) we oversample textual reasoning paths from a pretrained MLLM; (2) we apply an LLM-based parser to decompose each path into discrete visual queries and retrieval steps; and (3) we ground each visual reference by aligning it with a bounding box in the input image. Representative examples are provided in the Appendix.

**Constructing base reasoning traces.** As a seed to our grounded corpus, we adopt the training set of TVC (Sun et al., 2025), which consists of reasoning traces generated from the QvQ model (Qwen Team, 2024). The dataset encompasses nine distinct problem domains: Charts, Documents, Geometry, IQ Tests, Medical Imagery, Natural Scenes, Science Diagrams, Synthetic Images, and Tables.

**Decomposing reasoning traces into visual reference steps.** We extract visual grounding cues from text-based reasoning traces using a strong off-the-shelf LLM (Gemini-2.0-flash (Google, 2025)). The model rewrites each reference to visual content as a *detect* call, which takes a short natural language description and returns the corresponding image region. Retrieved objects are cached and assigned symbolic identifiers `<objX>` in order of appearance. In addition, the LLM generates a key–value list of visual components, with each key serving as a unique, descriptive grounding reference for later steps. We construct domain-specific few-shot prompts to guide this process, with prompt templates detailed in the Appendix. Finally, we post-process the LLM outputs to discard failure cases, including mismatches between referenced and retrieved objects, non-unique object labels, insufficient object count ($\leq 2$), and ill-formed reasoning. After filtering, $\sim 82\%$ samples are retained.

**Grounding visual references to image regions.** Visual grounding is challenging in multimodal reasoning tasks, since they often involve domains beyond natural images (*e.g.* charts, geometry, medical scans). Existing grounding models perform poorly on such inputs (Steiner et al., 2024; Xiao et al., 2024). Moreover, these tasks frequently require grounding abstract or symbolic cues (*e.g.* angle ABC), which lie outside the training scope of current models.

To exploit the implicit visual grounding behavior in MLLMs, we build on a visual grounding model Qwen2.5-VL (Bai et al., 2025). However, rather than relying on its coordinate generation interface, we estimate the model's visual focus using a relative attention mechanism inspired by Zhang et al. (2025), in order to better handle non-natural domains and symbolic cues. Specifically, we extract cross-attention maps from the grounding query to all image patches, with and without grounding prompts. The ratio of conditional to marginal attention yields a normalized map that highlights semantically

Table 1: **Results on multimodal mathematical reasoning tasks.** MathVision results are both reported for the mini and full subsets to include more baseline scores.

| Model | Size | Reasoning Only | MathVista mini | MathVision mini | MathVision full | MathVerse mini | Average mini | Average full |
|---|---|---|---|---|---|---|---|---|
| Qwen2-VL Wang et al. (2024c) | 7B | ✗ | 60.9 | - | 16.3 | 24.6 | - | 20.5 |
| Qwen2-VL Wang et al. (2024c) | 72B | ✗ | 69.7 | - | 26.6 | 36.2 | - | 31.4 |
| Qwen2.5-VL Bai et al. (2025) | 7B | ✗ | 67.8 | 23.6 | - | 44.5 | 45.3 | - |
| Qwen2.5-VL Bai et al. (2025) | 72B | ✗ | 74.8 | 39.8 | - | 57.6 | 57.4 | - |
| InternVL2.5 Chen et al. (2025) | 8B | ✗ | 64.4 | 22.0 | 19.7 | 39.5 | 41.9 | 29.6 |
| InternVL2.5 Chen et al. (2025) | 78B | ✗ | 72.3 | 34.9 | 32.2 | 51.7 | 53.0 | 42.0 |
| GPT-4o Hurst et al. (2024) | - | ✗ | 63.8 | - | 30.4 | 50.2 | - | 40.3 |
| LLaVa-CoT Xu et al. (2025) | 11B | ✓ | 54.8 | 16.3 | - | 33.9 | 35.0 | - |
| Mulberry Yao et al. (2024) | 7B | ✓ | 63.1 | - | - | 39.6 | - | - |
| TVC Sun et al. (2025) | 7B | ✓ | 68.1 | - | 22.7 | 38.9 | - | 30.8 |
| TVC Sun et al. (2025) | 72B | ✓ | 72.2 | - | 41.9 | 48.8 | - | 45.4 |
| QVQ-72B-preview Qwen Team (2024) | 72B | ✓ | 71.4 | 35.9 | - | 41.5 | 49.6 | - |
| Base (Qwen2.5-VL) | 7B | ✗ | 67.8 | 23.6 | - | 44.5 | 45.3 | - |
| Text-Only (TVC) | 7B | ✓ | 68.1 | - | 22.7 | 38.9 | - | 30.8 |
| **Ours** | 7B | ✓ | **68.6** | **34.5** | **28.1** | **48.6** | **50.6** | **38.4** |
| ↳ Inference w/o Pointing | 7B | ✓ | 60.0 | 25.3 | 23.7 | 33.6 | 39.6 | 28.7 |

meaningful regions while suppressing low-level register effects. A heuristic post-processing step then converts this soft mask into discrete bounding boxes, after which malformed or invalid boxes are discarded. This produces a curated training set of $\sim 300K$ grounded examples.

## 5 IMPLEMENTATION DETAILS

**Preprocessing.** Given a multimodal input consisting of interleaved images, text, and bounding box annotations for visual references, we first convert each image into a flattened sequence of image patches, following the patchification protocol used by the backbone model (e.g., Qwen2.5-VL (Bai et al., 2025)). Each bounding box is then transformed into a corresponding sequence of pointer tokens (e.g., `<ptr4>,...,<ptr32>`), where each token refers to an enclosed image patch. These pointer tokens are appended to the tokenizer vocabulary but do not modify the model's original embedding table or generation head. During preprocessing, the embeddings for pointer tokens are replaced with the corresponding image patch embeddings prior to the transformer layers. The final input is thus a unified sequence of text tokens, pointer tokens, and image patch embeddings.

**Model.** Our **v1** architecture is designed to extend a broad range of multimodal language model (MLLM) backbones; for empirical validation, we instantiate it on Qwen-2.5-VL. Architecturally, we introduce only two lightweight linear layers atop the original model: a pointing query head $L_q \in \mathbb{R}^{D \times D}$ and a pointing key head $L_k \in \mathbb{R}^{D \times D}$, where $D$ denotes the latent dimensionality of the MLLM. Both heads are initialized as identity matrices scaled by $1/\sqrt{D}$, ensuring that their influence on the model's initial output distribution remains minimal. This initialization strategy is particularly effective given the structure of our task: the pretrained backbone already produces meaningful generative likelihoods $P_{\text{gen}}$, and the pointing mechanism selects at most a single position per timestep from the pointing distribution $P_{\text{ptr}}$. As a result, the added modules integrate smoothly during early training and do not induce catastrophic forgetting.

**Training.** Given the dual-nature output space comprising a generative vocabulary $V$ and a pointing reference set $K$, we incorporate z-loss regularization to stabilize the softmax partition function, following Chowdhery et al. (2023); Wortsman et al. (2023); Team (2024). Specifically, we regularize the log-partition function $Z = \sum_j e^{x_j}$ in the softmax $\sigma(x)_i = e^{x_i}/Z$ by introducing a z-loss term $\mathcal{L}_z = \lambda \log \bar{Z}$, where $\lambda = 10^{-5}$. To reduce computational overhead, we approximate $Z$ using a top-$k = 40$ partition function $\bar{Z} = \sum_{j \in \text{TopK}(x)} e^{x_j}$. This approximation enables efficient and numerically stable training in large-output-space settings. Further details are in Appendix A.

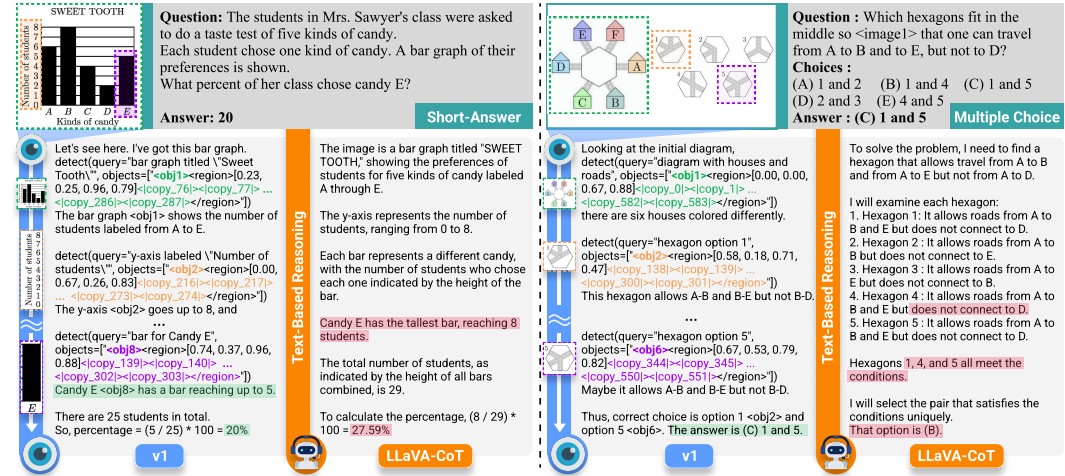

Figure 4: **Qualitative comparison on MathVision. v1**'s dynamic grounding helps to solve both bar graph and spatial reasoning tasks, while LLaVA-CoT misinterprets visual content in both cases.

**Inference.** At each decoding step to generate token $x_t$, **v1** utilizes two additional caches: (1) keys, given by hidden features $L_k(c)$ corresponding to image patch positions for computing the pointing logits $\text{logit}_{\text{ptr}}$, and (2) values, the input feature sequences of the associated image patches. These are essential for enabling the pointing and copying mechanism during inference. We implement the additional caches by extending the key-value attention caching interface of the HuggingFace (Wolf et al., 2020) Transformers library. The memory overhead is minimal compared to the standard attention cache and can be realized as an auxiliary attention layer with empty parameters serving solely as a cache. We plan to release the implementation with the codebase.

## 6 EMPIRICAL RESULTS

### 6.1 DOWNSTREAM EVALUATION ON MULTIMODAL REASONING BENCHMARKS

**Setup.** We use three representative multimodal mathematical reasoning benchmarks: MathVista (mini) (Lu et al., 2024), MathVision (mini/full) (Wang et al., 2024b), and MathVerse (mini) (Zhang et al., 2024a). Following prior work (Duan et al., 2024), we use GPTEval (Liu et al., 2023b) to compute accuracy while accounting for the formatting inconsistencies.

We compare our method against both general-purpose and reasoning-specialized MLLMs. General MLLMs include Qwen2-VL (Wang et al., 2024c) and Qwen2.5-VL (Bai et al., 2025) at both 7B and 72B scales, as well as InternVL2.5 (Chen et al., 2025) at 8B and 78B. We also include GPT-4o (Hurst et al., 2024) as a high-performing proprietary baseline. For reasoning-oriented models, we evaluate LLaVa-CoT-11B (Xu et al., 2025), Mulberry-7B (Yao et al., 2024), TVC-7B and 72B (Sun et al., 2025), and QVQ-72B-preview (Qwen Team, 2024).

**Results.** Quantitative results are presented in Table 1. Our approach yields substantial performance improvements over baseline models. Among 7B-scale models, **v1** with full pointing capability consistently outperforms both general-purpose and reasoning-specialized baselines. Notably, despite its smaller size, our 7B model narrows the performance gap with several 72B-scale models. The gains are particularly pronounced on MathVision, a benchmark known for its higher complexity and stronger demand for grounded reasoning in MLLMs.

### 6.2 FURTHER ANALYSIS

**Qualitative results.** Figure 4 shows a qualitative comparison between our method and LLaVA-CoT (Xu et al., 2025) as a baseline. In both the short-answer (left) and multiple-choice (right)

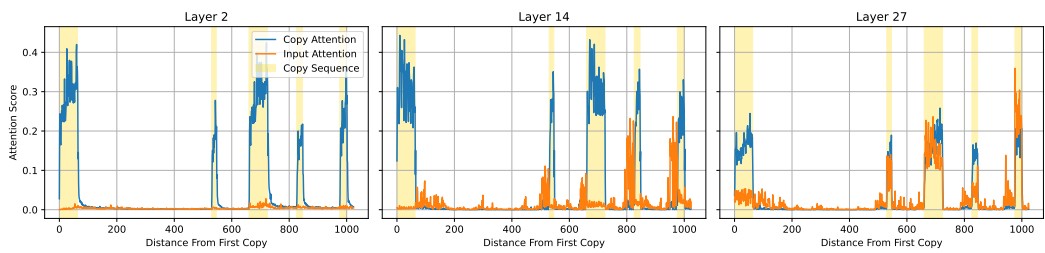

Figure 5: **Comparison of attention to copy tokens vs. original visual tokens.** Layer-wise sum of attention scores directed to copy tokens and their corresponding original visual input tokens from a **v1** output on a MathVision example. Copy token intervals are highlighted in yellow.

examples from MathVision, our **v1** demonstrates explicit visual grounding through pointer-based detection and selective copying of relevant image regions.

For the bar graph example, **v1** accurately identifies the bar corresponding to Candy E and computes the correct percentage based on the total count, while LLaVA-CoT misidentifies the tallest bar and overestimates the result. In the hexagon pathfinding task, **v1** correctly reasons over spatial connectivity by attending to the structural differences in the options, whereas LLaVA-CoT fails to filter invalid candidates and outputs the wrong answer. These examples highlight how active visual reference via pointing enables more precise and interpretable reasoning compared to text-only chain-of-thought approaches.

**Ablation study.** We conduct an ablation study, summarized in Table 2, to isolate the contributions of individual components in **v1**, with a focus on the impact of the proposed point-and-copy mechanism. We evaluate three ablated variants: (1) *Backbone*, the pretrained Qwen2.5-VL-7B model without any task-specific finetuning; (2) *Coordinate-Only*, which is trained on **v1g** using bounding box coordinates in place of pointer supervision; and (3) *Ours w/o Pointing*, which disables the pointing mechanism at inference time.

Table 2: **Ablation on MathVision testmini** to gauge the impact of dynamic visual reference.

| Variant | Train | Infer | Score |
|---|---|---|---|
| Backbone | ✗ | ✗ | 23.6 |
| + Coord-Only | ✗ | ✗ | 31.9 |
| Ours w/o Pointing | ✓ | ✗ | 25.3 |
| Ours | ✓ | ✓ | 34.5 |

The results indicate that the ability to actively retrieve and incorporate relevant visual tokens via pointing is critical for achieving strong performance on complex multimodal reasoning tasks.

**How does v1 utilize pointed visual regions?** We analyze how **v1** internally uses the visual regions retrieved via the point-and-copy mechanism. As shown in Figure 5, we compare the total *Input Attention* (attention to the original visual tokens) and *Copy Attention* (attention to the copied tokens) during generation after the first copy operation.

Our analysis reveals a coherent sequence of behaviors. First, attention to the original image tokens increases immediately before copying, indicating a localization step in which the model identifies where the relevant information resides. Second, immediately after copying, intermediate layers (e.g., layers 2 and 14) show a strong dominance of copy attention. This reflects an active post-retrieval processing phase where **v1** selectively emphasizes informative subcomponents of the retrieved region. Third, when attention patterns are averaged across layers, copied tokens consistently receive higher weight than the original image tokens, suggesting that once a region is copied into the language context, it becomes a stable and easily accessible reference. Finally, in higher layers (e.g., layer 27), attention to input and copied tokens becomes more balanced, which may correspond to a late-stage integration step in which the retrieved region is reconciled with the broader image context and used for planning subsequent reasoning.

Together, these observations show that **v1** uses pointed visual regions in a structured manner, transitioning from localization, to focused processing, and ultimately to high-level integration.

## 7 CONCLUSION

We introduced **v1**, a lightweight extension that enables MLLMs to actively revisit input images through a point-and-copy mechanism. To train it, we constructed **v1g**, a dataset of 300K multimodal reasoning traces with fine-grained visual grounding. Empirical results on established multimodal mathematical reasoning benchmarks demonstrate that **v1** significantly improves performance, particularly on tasks requiring grounded, step-by-step visual reasoning. We hope this work encourages further exploration of alternative methods for dynamic visual access as a core component of multimodal reasoning.

Looking forward, **v1**'s copy-and-method mechanism can be applied across modalities beyond text, such as speech and video. It can also be extended to flexible region retrieval beyond rectangular bounding boxes for iterative segmentation. Finally, it opens opportunities for controllable generation, where explicit reference signals are injected into the decoding process to constrain token selection and guide outputs toward designated regions.

### ETHICS STATEMENT

**v1** is designed for multimodal mathematical reasoning, a domain with minimal risk of direct societal harm. Nonetheless, we acknowledge that **v1** may inherit biases from its pretrained backbone (Qwen-2.5-VL). Our training dataset, **v1g**, consists solely of multimodal reasoning problems and was constructed as a re-annotation of an existing dataset (Sun et al., 2025), thereby minimizing potential privacy and licensing concerns. All human annotations involved in this work were performed by members of the research group, thereby avoiding potential ethical concerns associated with external crowd-sourced annotation labor. No separate human-subject studies were conducted.

### REPRODUCIBILITY STATEMENT

To ensure reproducibility, we provide comprehensive implementation details and will release all necessary resources. The v1 architecture and pointing mechanism are specified in Section 4, with training configurations (hyperparameters, initialization, z-loss regularization) in Section 5. Our method extends Qwen2.5-VL-7B with two additional linear layers ($L_q$ and $L_k$). Experiments were conducted on 8 NVIDIA A100 GPUs using DeepSpeed. The v1g dataset construction pipeline is documented in Section 4.3, with the data generation prompt template in Table 5. Evaluation protocols using GPTEval on MathVista, MathVision, and MathVerse benchmarks are described in Section 6. Upon publication, we will release: (1) the full v1g dataset with grounding annotations, (2) model checkpoints, (3) training and inference code for HuggingFace Transformers, and (4) evaluation scripts. The anonymized code and data samples are included with this submission.

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

## OVERVIEW OF THE APPENDIX

This Appendix is structured as follows:

- Appendix A describes implementation details and resources used in the project;
- Appendix B discusses limitations and future directions;
- Appendix C provides details of the data generation process;
- Appendix D compares attention patterns of **v1**'s point-and-copy method with coordinate-based reference;
- Appendix E reports human evaluation results validating the grounding quality of our training dataset (**v1g**) and our model (**v1**);
- Appendix F details pseudo-code on the visual grounding pipeline we utilized in the data generation process;
- Appendix G presents additional qualitative results.

## A    IMPLEMENTATION DETAILS & RESOURCES

**Training Details**    All models are trained under uniform settings: a base learning rate of $3 \times 10^{-5}$, per-device batch size of 2, and gradient accumulation over 4 steps. We leverage DeepSpeed for distributed training across 8 NVIDIA A100 GPUs. Optimization uses AdamW with $\beta_1 = 0.9$, $\beta_2 = 0.95$, and training is performed for 5 epochs.

**Training Duration**    Our training schedule of 5 epochs follows the setup used for the original text-only reasoning trace dataset, which we extend to the grounded reasoning setup Sun et al. (2025). Because the reasoning traces contain substantially longer token sequences than typical MLLM data, shorter training runs produced unstable behaviors such as repetition and incomplete reasoning without a final answer. In contrast, the point-and-copy behavior required relatively little data and typically saturated within the first epoch. The longer schedule therefore reflects the requirements of the inherited reasoning-trace setup rather than the needs of the pointer mechanism itself.

**Large Language Model Usage.**    LLMs (ChatGPT, GPT-4/5 class) were employed to refine phrasing, improve clarity, and standardize style in sections of the manuscript, but all scientific ideas, experiments, and analyses were conceived, executed, and validated by the authors. LLMs were also used in a limited capacity to assist with literature discovery (e.g., surfacing related work for manual screening). All substantive content decisions, experiment design, and result interpretation remain entirely author-driven.

## B    LIMITATIONS AND FUTURE WORK

This work focuses on demonstrating the effectiveness of active visual reference in structured multi-modal reasoning via a simple point-and-copy mechanism. While **v1** shows strong performance in mathematical domains, several directions remain for broader applicability.

**Beyond mathematical domains.**    Extending **v1** to other settings (*e.g.* such as scientific diagrams, medical images, or visual commonsense) presents new challenges in representation and supervision. These domains often lack structured reasoning traces, making data collection more difficult. Since **v1g** relies on a pretrained text-only MLLM to seed reasoning, generalizing to less structured domains will require advances in decomposition, grounding, and alignment.

**Weak supervision and reinforcement learning.**    Recent work in inference-time scaling and alignment has shown the promise of reward-based learning for reasoning. Incorporating such methods into **v1** may enable more flexible and efficient visual retrieval strategies without dense supervision. We leave this exploration to future work due to current resource constraints.

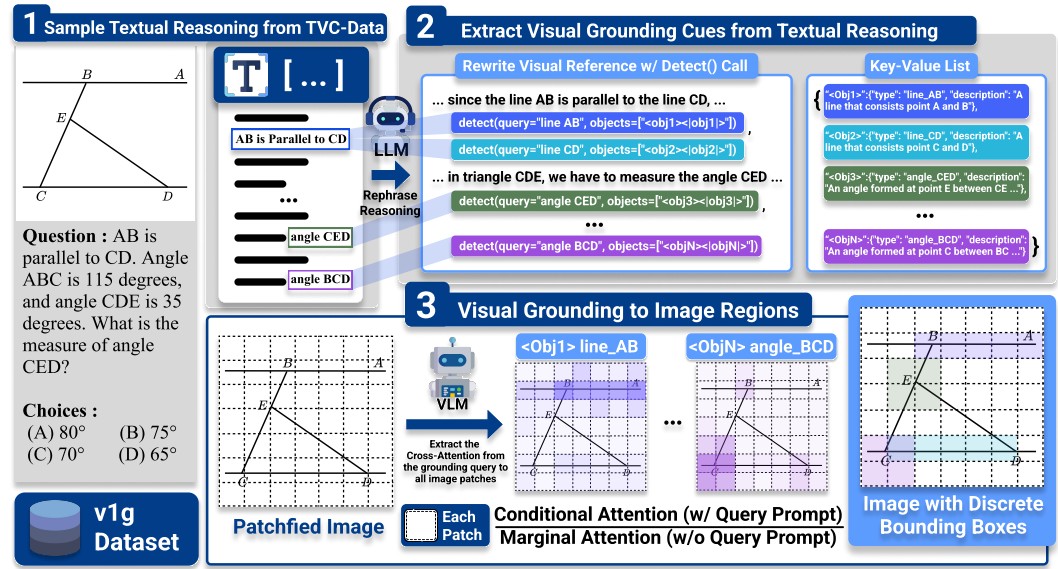

Figure 6: **v1g** dataset construction pipeline.

## C  DATA GENERATION DETAILS

Figure 6 illustrates the construction pipeline for our **v1g** dataset; each stage of this pipeline is described in detail in Section 4.3. The specific prompt template used to decompose text-based reasoning paths into visual queries (as outlined in our methodology in Section 4.3) is provided in Table 5.

## D  ATTENTION SCORE COMPARISON BETWEEN TEXT-BASED REASONING

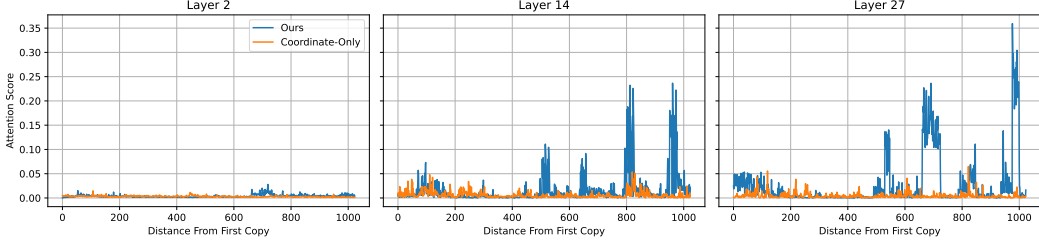

Figure 7: **Layer-wise Visual Attention Dynamics: v1 vs. *Coordinates-Only*.** Attention scores on visual inputs for **v1** (referenced input visual tokens during reasoning) versus the *Coordinates-Only* baseline (every input visual token), shown across layers (2, 14, and 27). The x-axis, "Distance From First Copy," tracks generation steps after **v1**'s initial copy operation.

Figure 7 presents a layer-wise comparison of these attention dynamics (using a MathVision example), plotting the sum of attention scores on original visual tokens against generation steps following **v1**'s first copy operation. For **v1**, these scores represent attention to specific visual tokens designated by its point-and-copy mechanism at each step. This targeted attention is pronounced and dynamic, particularly in intermediate and deeper layers (e.g., Layers 14 and 27), where scores fluctuate significantly, peaking at approximately 0.35, indicating active engagement with referenced visual information. In stark contrast, for the *Coordinates-Only* model, the sum of attention across all its original visual tokens (present in the context at each step) remains consistently low (generally below 0.05) and largely static across all layers. This comparison underscores how **v1**'s explicit pointing and copying mechanism enables more focused and substantial engagement with relevant visual

Table 3: Human evaluation of v1g dataset quality.

| Method | Metric | Avg | Majority | Fleiss' $\kappa$ | Agreement |
|---|---|---|---|---|---|
| Attention-based (Ours) | Correctness | 83.3% | 87.0% | 0.352 | Fair |
| | Comprehensive. | 55.0% | 56.0% | 0.582 | Moderate |
| | Tightness | 46.0% | 44.0% | 0.436 | Moderate |
| Grounding-DINO | Correctness | 29.3% | 30.0% | 0.711 | Substantial |
| | Comprehensive. | 47.3% | 49.0% | 0.906 | Almost Perfect |
| | Tightness | 19.0% | 18.0% | 0.675 | Substantial |

information during the reasoning process. The analysis window for both models commences from the generation step at which **v1** produced its first copy token, extending for a consistent number of subsequent steps.

# E  HUMAN EVALUATION OF GROUNDING QUALITY

## E.1  EVALUATION OF V1G DATASET QUALITY

To validate the quality of our automatically generated visual grounding annotations in the v1g dataset, we conducted a human evaluation comparing our attention-based grounding approach against GroundingDINO (Liu et al., 2024), a widely-used open-set object detector.

**Methodology.**  We randomly sampled 100 examples from the v1g dataset, each containing multiple bounding boxes. Three expert annotators independently evaluated each bounding box using binary classification on three criteria:

- **Correctness**: Whether the bounding box covers the intended object or region
- **Comprehensiveness**: Whether all relevant visual content is included within the box
- **Tightness**: Whether the box is well-fitted with minimal extraneous background

We report the average score across annotators, majority vote, and Fleiss' $\kappa$ to assess inter-annotator agreement. Agreement quality follows standard interpretations: Fair (0.21–0.40), Moderate (0.41–0.60), Substantial (0.61–0.80), and Almost Perfect (0.81–1.00).

**Results.**  As shown in Table 3, our attention-based grounding method substantially outperforms GroundingDINO on correctness (83.3% vs. 29.3% average score), demonstrating superior capability in localizing semantically complex and context-dependent entities such as geometric elements (e.g., "angle ABC"), chart components (e.g., "bar for Grace"), and referring expressions (e.g., "the second figure"). While GroundingDINO achieves higher inter-annotator agreement, this primarily reflects consistent failure modes rather than quality, as evidenced by its low absolute performance.

## E.2  EVALUATION OF V1 POINTING ACCURACY

We additionally evaluated the pointing accuracy of our trained v1 model to assess how effectively it grounds visual references during inference.

**Methodology.**  Using the same evaluation protocol, we sampled 100 outputs from v1 on the MathVision dataset. Annotators evaluated whether the model's pointed regions (copied image tokens) correctly corresponded to the referenced objects in the reasoning trace. We added an **Appropriateness** criterion to assess whether the pointing action was contextually justified.

**Results.**  Table 4 demonstrates that v1 maintains high grounding quality during inference, achieving 82.7% correctness—comparable to the training data quality. The high appropriateness score (87.7%) indicates that the model learns to selectively invoke the pointing mechanism when dynamic visual reference is genuinely beneficial for reasoning.

Table 4: Human evaluation of v1 model pointing quality.

| Metric | Avg | Majority | Fleiss' $\kappa$ | Agreement |
|---|---|---|---|---|
| Correctness | 82.7% | 87.0% | 0.558 | Moderate |
| Comprehensiveness | 55.7% | 54.0% | 0.689 | Substantial |
| Tightness | 49.3% | 40.0% | 0.280 | Fair |
| Appropriateness | 87.7% | 90.0% | 0.599 | Moderate |

**Discussion.** The evaluation reveals that our attention-based grounding excels at capturing semantically rich visual references that are challenging for traditional object detectors. The moderate tightness scores across both methods reflect the inherent ambiguity in defining precise boundaries for abstract concepts (e.g., "angle 2" or "the second figure"), where multiple valid interpretations exist. The consistency between training data quality and model performance suggests that v1 successfully learns robust visual grounding capabilities from our automatically generated supervision.

## F  BOUNDING-BOX EXTRACTION FROM CROSS-ATTENTION

This section provides a high-level pseudocode description of our data annotation method for deriving bounding boxes from cross-attention in Qwen-2.5-VL.

---
**Algorithm 1:** Bounding-Box Extraction from Cross-Attention (High-Level)

---
**Input:** Image $I$, region description $T$
**Output:** Bounding box $b$ corresponding to $T$
**1. Prepare multimodal input.**
Concatenate $I$ with a static visual-grounding instruction prompt and feed it to Qwen2.5-VL.
**2. Extract attention with instruction.**
From the final decoding position, obtain the cross-attention map $A$ over image tokens. Use a
  predefined set of layers (selected empirically) and average across heads.
**3. Extract baseline attention.**
Remove the object name from the prompt, feed the modified prompt with $I$ to the model, and
  extract the corresponding attention map $A'$ using the same layers and averaging.
**4. Compute attention contrast.**
Compute the contrastive relevance for each image token:

$$R = \frac{A}{A'}.$$

**5. Derive bounding region.**
Identify the peak region in $R$. Sweep over multiple candidate crop ratios; for each ratio, form a
  bounding region around the peak. Select the bounding box maximizing contrast sharpness
  between inside and outside regions. Convert the selected region to image-coordinate bounding
  box $b$.
**6. Return.**
**return** $b$

---

## G  ADDITIONAL QUALITATIVE RESULTS

To further illustrate **v1**'s complex visual reasoning, this section provides additional qualitative examples, complementing Figure 4 from the main text. These examples highlight how **v1** leverages the point-and-copy mechanism.

Figure 8 demonstrates **v1** on a synthetic task (CLEVR-like) requiring object counting based on the query: "Subtract all red things, then subtract all tiny matte balls. How many objects are left?". **v1** first localizes objects using its pointer mechanism. It then sequentially reasons, identifying "red" objects before revisiting relevant items, like the "cyan sphere," to verify the combined "tiny" and "matte"

attributes through targeted attention. This process demonstrates **v1**'s capacity for precise attribute grounding and multi-step compositional reasoning enabled by the point-and-copy mechanism.

In Figure 9, **v1** tackles a chart comprehension task: determining if the "Dark Violet" data series has the minimum area under the curve. **v1** initially grounds key chart elements, using its pointer to isolate data series such as "Dark Violet," "Medium Mint," and "Dark Cyan." Later in its reasoning, it proactively revisits these series, performing a comparative analysis of their visual trajectories and relative y-axis values to infer their respective areas. Such selective re-focusing showcases its ability to perform nuanced comparisons within dense visual information.

These examples further affirm that **v1**'s architecture, by supporting active visual reference and precise grounding via its pointing mechanism, achieves robust, interpretable, and accurate multi-step visual reasoning.

**Question:**
Subtract all red things. Subtract all tiny matte balls. How many objects are left?

**Answer: 5**

**Image-Question Pair**

Let me see what objects are present. I need to identify all the objects first.
detect(query="yellow cube", objects=["**<obj1><region>[0.00, 0.00, 0.21, 0.55]<|copy_0|><|copy_1|> ... <|copy_45|><|copy_46|></region>**"])
detect(query="blue cylinder", objects=["**<obj2><region>[0.29, 0.20, 0.46, 0.45]<|copy_14|><|copy_15|> ... <|copy_48|><|copy_49|></region>**"])
detect(query="green cylinder", objects=["**<obj3><region>[0.58, 0.20, 0.75, 0.50]<|copy_17|><|copy_18|> ... <|copy_50|><|copy_51|><|copy_52|></region>**"])
detect(query="purple cylinder", objects=["**<obj4><region>[0.75, 0.45, 0.96, 0.65]<|copy_52|><|copy_53|> ... <|copy_64|><|copy_65|></region>**"])
detect(query="small green cylinder", objects=["**<obj5><region>[0.12, 0.45, 0.29, 0.75]<|copy_45|><|copy_46|> ... <|copy_68|><|copy_69|></region>**"])
detect(query="red sphere", objects=["**<obj6><region>[0.46, 0.30, 0.54, 0.45]<|copy_27|><|copy_38|><|copy_49|></region>**"])
detect(query="cyan sphere", objects=**["**<obj7><region>[0.33, 0.65, 0.46, 0.85]<|copy_58|><|copy_59|> ... <|copy_81|><|copy_82|></region>**"])

The task is to subtract all red things and all tiny matte balls.
Let me consider the order: first, subtract all red things, then subtract all tiny matte balls.
Step 1: Subtract all red things. There's only one red sphere <obj6>. So, remove that.
Remaining objects: 1. Yellow cube <obj1>  2. Blue cylinder <obj2>  3. Green cylinder <obj3>  4. Purple cylinder <obj4>  5. Small green cylinder <obj5>  6. Cyan sphere <obj7>
Step 2: Subtract all tiny matte balls. Assuming the cyan sphere <obj7> is matte, it should be removed. So, remove the cyan sphere <obj7>.
Remaining objects: 1. Yellow cube <obj1>  2. Blue cylinder <obj2>  3. Green cylinder <obj3>  4. Purple cylinder <obj4>  5. Small green cylinder <obj5>
That's five objects.

Wait, but the problem might be considering only the red sphere <obj6> as matte, and the cyan sphere <obj7> is also matte. But since it's not specified, I'll assume both are matte.

Alternatively, perhaps "tiny matte balls" refer only to the cyan sphere <obj7>, and the red sphere <obj6> is not considered a "ball" or something. But the problem mentions "tiny matte balls," and both spheres are small and matte, so probably both are considered tiny matte balls. Therefore, after subtracting the red sphere <obj6> and the cyan sphere <obj7>, we're left with five objects: yellow cube <obj1>, blue cylinder <obj2>, green cylinder <obj3>, purple cylinder <obj4>, and small green cylinder <obj5>.

But wait, the user's message seems to indicate there are six objects initially, but lists seven, which is confusing. Maybe there's a mistake in counting. Alternatively, perhaps the small green cylinder <obj5> is not present, and there are only six objects: Yellow cube <obj1>, Blue cylinder <obj2>, Red sphere <obj6>, Green cylinder <obj3>, Purple cylinder <obj4>, Cyan sphere <obj7>

In this case, after subtracting the red sphere <obj6> and the cyan sphere <obj7><region>[0.33, 0.65, 0.46, 0.85]<|copy_58|><|copy_59|> ... <|copy_81|><|copy_82|></region>, we're left with four objects: yellow cube <obj1>, blue cylinder <obj2>, green cylinder <obj3>, purple cylinder <obj4>.

But the user mentioned six objects, including the small green cylinder <obj5>, so I'll stick with that.

Alternatively, maybe the small green cylinder <obj5> is separate from the green cylinder <obj3>, making it seven distinct objects. Regardless, the key is to subtract the red sphere <obj6> and the cyan sphere <obj7>, assuming both are matte.
So, final answer should be 5 or 4, depending on whether the small green cylinder is considered separate or part of the green cylinder. But since the user specified seven objects, including small green cylinder <obj5>, the answer should be 5. **Final Answer**\[ \boxed{5} \]

Figure 8: Qualitative example of **v1** tackling an attribute-based counting task in a synthetic domain.

**Question:**
Does Dark Violet have the minimum area under the curve?

**Choices:**
(A) yes   (B) no

**Image-Question Pair**

Alright, I've got this graph to analyze. First, I need to detect the graph.
detect(query="the graph", objects=["**<obj1><region>[0.00, 0.00, 1.00, 1.00]<|copy_0|><|copy_1|> ... <|copy_310|><|copy_311|></region>**"])
There are five different data series here, each represented by a different color: Dark Violet, Hot Pink, Medium Mint, Dark Cyan, and Yellow Green. I need to detect each data series.
detect(query="Dark Violet data series", objects=["**<obj2><region>[0.12, 0.77, 0.79, 0.93]<|copy_219|><|copy_220|> ... <|copy_306|><|copy_307|></region>**"])
detect(query="Hot Pink data series", objects=["<obj3><region>[0.21, 0.23, 0.58, 0.46]<|copy_52|><|copy_53|> ... <|copy_157|><|copy_158|></region>"])
detect(query="Medium Mint data series", objects=["**<obj4><region>[0.12, 0.54, 0.54, 0.85]<|copy_147|><|copy_148|> ... <|copy_276|><|copy_277|></region>**"])
detect(query="Dark Cyan data series", objects=["**<obj5><region>[0.25, 0.15, 0.54, 0.38]<|copy_30|><|copy_31|> ... <|copy_108|><|copy_109|></region>**"])
detect(query="Yellow Green data series", objects=["**<obj6><region>[0.08, 0.07, 0.79, 0.77]<|copy_1|><|copy_2|> ... <|copy_258|><|copy_259|></region>**"])

The y-axis is labeled "label" and ranges from 86 to 98, and the x-axis is numbered from 0 to 100. I need to detect the y-axis and x-axis.
detect(query="y-axis", objects=["<obj7><region>[0.00, 0.30, 0.04, 0.62]<|copy_72|><|copy_73|> ... <|copy_192|><|copy_193|></region>"])
detect(query="x-axis", objects=["<obj8><region>[0.33, 0.84, 0.50, 1.00]<|copy_247|><|copy_248|> ... <|copy_298|><|copy_299|></region>"])

Looking at the plot <obj1>, I can see that each of these has points scattered across the graph area.
Starting with Dark Violet <obj2>, which is purple. It seems like there are several purple points clustered towards the lower part of the graph <obj1>, around the 86 mark on the y-axis <obj7>. They don't seem to vary much in value, suggesting that this might be a relatively flat line or very little change over the x-axis <obj8> range.
Hot Pink <obj3>, which is light pink, shows a more dynamic pattern. The points start high around 98 and seem to decrease gradually as the x-axis <obj8> progresses. There might be some fluctuation, but overall, there's a downward trend.
Medium Mint <obj4>, the light greenish-blue, starts low around 86 and increases steadily, possibly following a linear pattern. By the end, it reaches up to about 92 or 93.
Dark Cyan <obj5>, the dark blue-green, appears to be quite variable. It starts around 96 and fluctuates up and down, sometimes dipping below 94 and rising above 96.
Yellow Green <obj6>, the yellowish-green, also shows variability, starting around 96 and fluctuating between about 92 and 96.

...

From the description, it seems that Dark Violet <obj2> is consistently around 86, with minimal variation. Comparatively, Medium Mint **<obj4><region>[0.12, 0.54, 0.54, 0.85]<|copy_147|><|copy_148|> ... <|copy_276|><|copy_277|></region>** starts at 86 and increases to about 92 or 93, so its average would be higher than Dark Violet's <obj2>. Similarly, Dark Cyan <obj5><region>[0.25, 0.15, 0.54, 0.38]<|copy_30|><|copy_31|> ... <|copy_258|><|copy_259|></region> have higher y-values, so their areas would be larger.

Hot Pink <obj3><region>[0.21, 0.23, 0.58, 0.46]<|copy_52|><|copy_53|> ... <|copy_157|><|copy_158|></region> starts high and decreases, but even at its lowest, it's around 90, which is still higher than Dark Violet's <obj2> average.
Therefore, based on this analysis, Dark Violet <obj2> indeed has the smallest area under the curve.
So, the answer should be (A) yes. **Final Answer**\[ \boxed{A} \]

Figure 9: Qualitative example of **v1** performing comparative reasoning on a chart comprehension task.

1134
1135
1136
1137
1138
1139
1140
1141
1142
1143
1144
1145
1146
1147
1148
1149
1150
1151
1152
1153
1154
1155
1156
1157
1158
1159
1160
1161
1162
1163
1164
1165
1166
1167
1168
1169
1170
1171
1172
1173
1174
1175
1176
1177
1178
1179
1180
1181
1182
1183
1184
1185
1186
1187

**Prompt for data generation**

You are given text-only reasoning for visual question answering.
Your task is to convert this text-only reasoning into visually grounded reasoning.
### STEP-BY-STEP INSTRUCTION
Please following these instructions step-by-step, imitating human visual reasoning behavior by:
1) Start from the beginning of the reasoning and read EACH sentence.
2) When you think you'd better to look the object or region, use detect() function
3) Format: 'detect(query="visual item that you want to find", objects=["<obj#>"])'
4) After detection, reference the visual element with '<obj#>' tags everytime you need to look it again immediately after mentioning the item.
5) Use NEW object numbers ('<obj1>', '<obj2>', '<obj3>'...) for EACH new detection.

### EXAMPLE:
Original text:
"Looking at the graph, I can see the function reaches its maximum at x = 3."

Corrected:
```
To answer the question, I need to look the graph.
detect(query="function graph", objects=["<obj1>"])
Looking at the graph <obj1>, I can see the function reaches its maximum at x = 3.
```
Later reference:
You can skip the <obj#> tag when you think you do not need to look it again.
```
The slope of the function becomes zero at this point on the graph.
```

### KEY REQUIREMENTS:
- Every item in lists MUST have its own 'detect()' statement
- Put 'detect()' statements on their own lines
- NEVER skip any visual element mentioned in the reasoning
- Start object numbering at 'obj1' and increase by 1 for each new object

### <OBJ#> REQUIREMENTS
- Visual element should be concrete, distinct, and explicit. Later you will localize the element based on the detect(). So make sure that the element not confusing.
- Use separate tags for each object (write "between the bus <obj1> and the car <obj2>" not "between <obj1 and obj2>").
- GOOD grounding: "I need to analyze this problem. detect(query="triangle", objects=["<obj1>"]) The triangle <obj1> has a right angle at vertex S."
- BAD grounding: "detect(query="triangle and rectangular", objects=["<obj1>"]) in the diagram, there are the triangle and rectangular has a right angle." (referring to non-atomic element)
- BAD grounding: "detect(query="region", objects=["<obj1>"]) The triangle <obj1> has a right angle." (referring to ambiguous element)

After completing the reasoning, list all objects detected:
{
"obj1": {"type": "function_graph",
"description": "Graph of a function with maximum at x = 3"},
"obj2": {"type": "next_item",
"description": "Description of next item"}
}

- We will localize the element with the open-world detector based on the descrip-

tion, so make sure to include well-described full self-contained description enough to uniquely identify the object.

### FINAL FORMAT:
{
"reasoning": "Your fully visually-grounded reasoning text",
"obj_list": "Your JSON object list"
}

Now, strictly following the instruction and the example, please provide the object list and visually grounded reasoning for the following prompt and reasoning:

### Example
Original Conversation

HUMAN:
[few_shot_question]

GPT:
[few_shot_answer]

### Visually Grounded Reasoning

GPT:
[few_shot_reasoning]

### Object List:

[few_shot_objects]

Now, given the conversation, please convert GPT's text-only reasoning into visually grounded reasoning

Original Conversation:

HUMAN:
[question]

GPT:
[answer]

### Visually Grounded Reasoning:

Table 5: Prompt used for converting textual reasoning to grounded reasoning annotations in **v1g** data generation process

