# OpenReview forum: "v1: Learning to Point Visual Tokens for Multimodal Grounded Reasoning"
_ICLR.cc/2026/Conference — Submitted to ICLR 2026_

### Official Review · Reviewer_db5f · 2025-10-27

**Soundness:** 3
**Presentation:** 2
**Contribution:** 3
**Rating:** 4
**Confidence:** 3

**Summary:**

The authors propose a lightweight extension that explicit re-access region in the input images to avoid visual groundting decay, which helps MLLM to do complex visual reasoning tasks. Their experiments show good performance on visual-math problems and they also claim the training data will be released.

**Strengths:**

1. The proposed methods sound reasonable and it is a lightweight extension that can be easily used on many MLLM.
2. The section 3 provides a solid support for the motivation of the methods.
3. The authors provide dataset to finetune its model. Also, the generation method is interesting and inspired.

**Weaknesses:**

1. The Figure 1 seems to sometimes invisible or incorrectly rendering. Sometimes, the region 2z-15 is too large and covering surrounding text. Please test your image in different devices.
2. Table 1 is too far from the related text.
3. [**MAJOR**] The ablation study in Table 2 is not significant. Specifically, the coord-method improve the backbone by 8.3% while your pointing method improve the w/o pointer version by 9.2%. The difference is not significant to show the benefits of your pointing comparing to the coord.

If the point 3 has been solved, I will raise my score to 6. If some of the following questions can be answered reasonably, I will further raise my score.

**Questions:**

1. One of the benifit of the method is that it is a lightweight extension and can be used with many existing pretrained MLLM. Another way that has been used in GPT-4o is to let MLLM to write some code to operate the images and put the new images into the context again. The latter one shows better flexible because it can do more on the original images rather than only point the original region. Can the authors provide more comparison between these two methods?
2. Also, the authors claim the "coordinate" method will "fail in cases where relevant visual cues are abstract or not spatially localized". Can the authors provide some experiments to show the better localizing ability of the point method?
3. The experiments in section 3 can be better explained. First, it is possible that the useful information has been extracted and reprensented in the text in the early stage of the generation, so that it is not necessary to revisit the images. Then, in Figure 3(b), 0.8 is not a significant low value, which raise a question that whether the decay will influnce the performance. I believe the author should provide more evidence about the relation between the decay and the performance.
4. Using cross-attention to ground the region is an interesting idea but it requires some experiments to support its effectiveness. Is there any post-examination or double-check to support its alignment with human intuition or "correct" useful region?

---

> ### Author Response · Authors · 2025-11-22
>
> We sincerely thank the reviewer for the constructive and insightful feedback. We are encouraged by your recognition that our method is a reasonable and lightweight extension, supported by solid motivation and an inspired data generation process.
> We address the following feedback below:
>
> ---
>
> ## **Weakness 1. Figure errors**
>
> We thank the reviewer for notifying us about this. We tracked down the source of this problem (it was sometimes not well-rendered in Safari browsers due to bugs in Figma) and will fix this by ensuring all figures are rasterized in the final revision. We deeply appreciate your patience in reviewing the manuscript despite this technical issue.
>
> ---
>
> ## **Weakness 2. Table-Content alignment**
>
> We thank the reviewer for notifying us about this. We will place the table closer to the corresponding text.
>
> ---
>
> ## **Weakness 3. Clarification on results in Table 2**
>
>
> The differences highlighted in the review arise from comparing mismatched references. The appropriate comparison uses the same base model as the shared backbone. Relative to this base:
> coord → +8.3%
>
>
> ours → +10.9%
>
>
> The remaining 2.6% therefore isolates the contribution of our pointing-based grounding, while the shared 8.3% reflects the benefit of grounded reasoning itself. The 9.2% noted by the reviewer compares our method to w/o Pointer, whereas coord is compared to the base, which mixes two different reference points and conflates grounded-reasoning effects with pointing effects.
> For clarity, w/o Pointer is not a trained baseline. It has the same weights as our v1 model, but we disable point-and-copy at inference. Its sole purpose is to confirm that the gain does not come from unrelated artifacts (e.g., different training dynamics), and it should not be interpreted as a competitive baseline for delta comparison.
> Our goal is twofold:
> demonstrating the value of grounded reasoning with our grounding annotation, and
>
>
> implementing this capability effectively through pointing.
>
>
> We will revise the table formatting to make these intended comparisons explicit.
>
>
> ---
>
> ## **Question 1. Comparison with tool calling**
>
> We clarify the distinction along two dimensions: conceptual categorization and implementation behavior.
>
> **Concept: Internalized skill vs. external tool**
>
>
> Tool calling requires models to convert visual references into structured code, but prior work shows that MLLMs still struggle to link indicated regions to code representations, even for simple geometric constructs [1]. This difficulty increases given recent findings that MLLMs often fail to recognize basic polygon shapes or count sides reliably [2]. If the model cannot robustly identify polygons, mapping regions to coordinate lists or code becomes even harder.
>  Our method avoids these issues because it operates entirely on similarity signals inside the model’s own representation space, keeping grounding aligned with internal reasoning rather than external code generation.
>
> **Implementation: Zooming behavior**
>
>
> While our method offers a weaker degree of freedom than full code execution, how these freedoms are used in practice is what matters. The GPT O3 report shows that 7 of 8 visual reasoning examples rely only on zooming or rotation, despite having access to full tool calling [3]. This suggests that many cases are handled with simple manipulations even when more general tools are available. Our mechanism focuses on this reliable regime and matches the interaction patterns that current MLLMs most effectively use.
>
>
> ---
>
> ## **Question 2. Experiment: coordinate-based vs. v1 grounding**
>
> We appreciate the suggestion and agree that a controlled comparison would strengthen the paper. We are training both a coordinate-based baseline and our v1-style grounding mechanism on standard grounding datasets. This requires substantial time, so we will update the response once results are ready or include the full results in the revision if the experiment finishes later.
>
> We also have indirect evidence from our annotation pipeline. We initially attempted to use the coordinate-based interface of Qwen-2.5-VL. Although it worked well on natural images, it failed entirely on mathematical plots and charts. In contrast, attention-based grounding from the same model generalized reliably in these settings, which motivated our design choice.
>
> Also, prior work shows that feature-based methods handle irregular regions more effectively than explicit coordinates [4,5]. This reflects a general limitation of coordinate interfaces beyond simple boxes. While we do not explore complex shapes due to annotation difficulty, our attention-based grounding is naturally compatible with such extensions.

---

> > ### Author Response · Authors · 2025-11-22
> >
> > ## **Question 3. Relation between decay and performance**
> >
> > We agree that attention decay is not inherently harmful; the problem is premature decay before sufficient visual information is extracted. We provide two supporting pieces of evidence:
> >
> > 1. Causal evidence.
> >
> > A prior study [6] demonstrated that simply re-inserting the input image during reasoning improves downstream performance, indicating that visual information is *not* fully extracted early in the chain. Table 2 from that work reports:
> >
> > | Method          | MathVista | MathVision | MathVerse |
> > | :---------------- | -------: | --------: | --------: |
> > | Baseline       | 63.5 | 19.8 | 31.6 |
> > | Visual revisit | 66.2 | 22.3 | 34.7 |
> >
> > 2. Qualitative evidence.
> >
> > We manually inspected 100 random trajectories generated by TVC on MathVision. Only 29  outputs correctly captured all relevant visual details within the first 500 tokens. Machine-based classification (Gemini-2.5-Flash) produced an identical rate. These results show essential visual details are not fully captured before attention decay happens.
> >
> > We plan to expand this study across more long reasoning models in the final draft.
> >
> >
> > ---
> >
> > ## **Question 3-1. Clarification on Figure 3(b)**
> >
> > We apologize for our lack of explanation. Figure 3(b) reports a *relative* attention measure, defined as $R = \frac{\frac{1}{|S|}\sum_{i \in S} A_i}{\frac{1}{|I|}\sum_{j \in I} A_j}$, where $A$ denotes attention scores, $S =$ set of salient-region patches and $I =$ set of all image patches.
> >
> > Therefore, a value of 0.8 does **not** indicate strong grounding. It means that a patch inside the salient region receives **less** attention than an average patch in the image. Hence, the model distributes attention more heavily toward non-salient regions rather than the evidence-relevant area.
> >
> > Prior work [6] also reports that sharper, more concentrated attention patterns correlate with higher model accuracy.
> >
> > ---
> >
> > ## **Question 4. Quality check on the visual grounding capacity**
> >
> > Since there is no canonical ground-truth set of grounding regions for each problem, we manually evaluate the detect-call behavior. Using v1’s reasoning trajectories on the MathVision dataset, we randomly sample 100 trajectories and assess each detect call along four criteria:
> >
> > - **Appropriateness:** whether the predicted box is relevant to the reasoning step.
> > - **Correctness:** whether the box identifies the intended object or region.
> > - **Comprehensiveness:** whether the box covers all relevant parts of the target.
> > - **Tightness:** whether the box minimizes background and is tightly aligned.
> >
> > Three annotators independently evaluate each sample for a binary decision. The results show that v1 reliably produces appropriate and correct grounding, while the spatial extent of the boxes is less precise, as reflected in lower comprehensiveness and tightness scores.
> >
> > | Category          | Maj Vote | Avg Score | Fleiss' $\kappa$ |
> > |:------------------|---------:|----------:|-----------------:|
> > | appropriateness   | 90.0%    | 87.7%     | 0.599            |
> > | comprehensiveness | 54.0%    | 55.7%     | 0.689            |
> > | correctness       | 87.0%    | 82.7%     | 0.558            |
> > | tightness         | 40.0%    | 49.3%     | 0.280            |
> >
> > ---
> >
> > ### **References**
> >
> >
> > [1] ChartM3, ACM Multimedia 2025 (https://arxiv.org/pdf/2507.21167v1)
> >
> > [2] Shape-Blind, ACL Findings 2025 (https://arxiv.org/pdf/2502.15969)
> >
> > [3] https://openai.com/index/thinking-with-images/
> >
> > [4] Text4Seg, ICLR 2025 (https://arxiv.org/pdf/2410.09855)
> >
> > [5] LISA, CVPR 2024 (https://arxiv.org/pdf/2308.00692)
> >
> > [6] AdaptVis, ICML 2025 (https://arxiv.org/pdf/2503.01773)

---

> > > ### Comment · Reviewer_db5f · 2025-11-25
> > >
> > > **Weakness 3**:
> > >     Yes, the concern is just that the main improvement comes from the coord method (8.3%), instead of your pointing method (2.6%). The comparison weakens the contribution.
> > >
> > > **Question 1**:
> > >     External tool is not worse than internal reasoning. If the authors believe it is true, they should provide some comparing experiments.
> > >
> > > **Question 2**:
> > >     Yes, I hope the new experiments can provide more evidence about the better localizing ability of the point method. I am not sure whether the "complex shapes" instead of simple boxes are necessary in practice.
> > >
> > > **Question 3**:
> > >     The evidence one is too indirect for me because re-inserting is another thing. I expect the new experiments in your final draft. Notice that you can provide modified version during the rebuttal.
> > >
> > > **Question 4**:
> > >     Yes, I think the crowd experiments are good to support the effectiveness of the cross-attention grounding. But I believe it is better to provide a baseline.

---

> > > > ### Author Response · Authors · 2025-12-02
> > > >
> > > > ## **Weakness 3: Significance of the gain**
> > > >
> > > >
> > > > We respectfully suggest that focusing solely on this delta narrows the scope of our contributions. Our paper presents a **holistic framework**, where the dataset (v1g) and the modeling approach (v1) are complementary contributions:
> > > > **The +8.3% gain** validates the high quality of our dataset and **the efficacy of the grounded reasoning objective itself**. This confirms a primary claim of our paper: that explicit grounding fundamentally improves reasoning.
> > > > **The additional +2.6% gain** validates our **Point-and-Copy architecture** as a superior implementation of this capacity.
> > > >
> > > >
> > > > We emphasize that Table 2 is an **ablation study** designed to disentangle these factors. The strong performance of the coordinate baseline (which also uses our data) should not detract from the paper's novelty; rather, it underscores the value of the grounded reasoning framework we have introduced.
> > > >
> > > >
> > > > ---
> > > >
> > > >
> > > > ## **Question 1: Comparison with Tool/Code Use**
> > > >
> > > > We are not arguing that internal reasoning is "better" than tools in all cases, but that they have fundamentally **different properties**. As requested, we highlighted these distinctions: Tool-use introduces dependencies on external executors, whereas our method internalizes the skill, keeping the reasoning process end-to-end differentiable and contained within the model's latent space.
> > > >
> > > >
> > > > ---
> > > >
> > > >
> > > > ## **Question 2: Necessity of pointing for "complex shapes"**
> > > >
> > > >
> > > > This distinction fundamentally parallels **segmentation vs. detection**. Coordinate methods are restricted to rectangular boxes, which fail to precisely capture the non-rectangular regions ubiquitous in charts and geometry (e.g., diagonal lines, curves). As noted in our initial response, prior work [4,5] confirms that feature-based methods handle such irregular regions far more effectively than explicit coordinates. Our pointing method operates on attention maps—effectively functioning as a segmentation mask—which allows it to isolate these complex shapes naturally, a flexibility that coordinate-based interfaces inherently lack.
> > > >
> > > >
> > > > ---
> > > >
> > > >
> > > > ## **Question 3: Evidence of Decay**
> > > >
> > > > We respectfully disagree that the re-insertion experiment is "indirect." It is **direct counterfactual evidence**. If the model had successfully retained the necessary visual information from the first pass, re-inserting the image would yield zero performance gain. The fact that re-accessing the image improves performance is proof that the information was lost (decayed) or insufficiently processed in the initial pass.
> > > >
> > > >
> > > > ---
> > > >
> > > >
> > > > ## **Question 4: Baselines for Human Evaluation**
> > > >
> > > > We thank the reviewer for recognizing the contribution of our additional human study. As our work proposes a **new framework**, there is no directly comparable prior baseline. For reference, we evaluated the closest related approach [1], which is concurrent with our work, under our math-reasoning setup (MathVision) and found that it was unable to produce even a single region due to the domain difference. We will summarize this results and update the draft accordingly.
> > > >
> > > >
> > > > ---
> > > >
> > > >
> > > >
> > > > ### **References**
> > > >
> > > >
> > > > [1] GRIT (https://arxiv.org/abs/2505.15879)

---

### Official Review · Reviewer_2iuN · 2025-10-30

**Soundness:** 2
**Presentation:** 3
**Contribution:** 3
**Rating:** 6
**Confidence:** 4

**Summary:**

The paper "v1: Learning to Point Visual Tokens for Multimodal Grounded Reasoning" proposes a novel mechanism to enhance multimodal large language models (MLLMs) by enabling them to re-access visual information during reasoning. Inspired by how humans repeatedly revisit visual stimuli while thinking, the authors design a point-and-copy module that allows the model to identify relevant image patches and inject their embeddings back into the reasoning stream.

To train this capability, they build v1g, a dataset containing 300K multimodal reasoning traces with interleaved visual grounding annotations. Experiments across several multimodal mathematical reasoning benchmarks show that v1 achieves strong performance gains over comparable baselines, demonstrating the potential of dynamic visual access for grounded reasoning.

**Strengths:**

### 1. Conceptually inspired and technically elegant:

The paper draws inspiration from human problem-solving processes and introduces a clever point-and-copy mechanism that allows re-referencing of visual regions without introducing additional vocabulary tokens.

### 2. Insightful analysis of visual grounding behavior:

The authors identify and analyze issues such as attention degradation and mismatch in visual token importance during long reasoning chains. This motivates the design of mechanisms that preserve visual grounding, making the argumentation coherent and compelling.

### 3. Strong empirical results and valuable dataset:

Through rigorous experiments, v1 demonstrates clear improvements on multimodal mathematical reasoning benchmarks. The proposed v1g dataset is likely to be a useful resource for future research on multimodal reasoning and grounding.

**Weaknesses:**

### 1. Dependence on external text-trace generation:

The SFT data pipeline relies on an external model (Gemini) to produce text-based traces. This is fine, but it remains unclear how reliably v1 can autonomously propose and execute such traces after training.

**Question:** Is there a quantitative evaluation of the success rate and reliability of detect-call usage during inference?

### 2. Lack of clarity in visual trace generation:

Section 4.3 states that v1’s visual traces are derived via heuristic post-processing of cross-attention maps, yet the details of this algorithm are not provided.

**Suggestion:** Include pseudo-code or a concise algorithmic description in the main text (possibly condensed from Appendix E) along with statistical analysis of the heuristic’s stability.

### 3. Ambiguity in model composition:

It is unclear whether the visual traces during inference are extracted by v1 itself or by a separate pretrained model (e.g., Qwen).

**Question:** How many models are actually involved in the inference loop, and are all functionalities integrated within v1 after training?

### 4. Limited interpretability discussion:

The ablation study on “How does v1 utilize pointed visual regions?” lacks depth.

**Suggestion:** A more detailed explanation would enhance understanding of how v1 internally uses the pointed regions to support reasoning.

### 5. Training procedure justification:

The choice of training for five epochs deviates from common MLLM practice (typically 1–2 epochs).

**Question:** Could the authors provide additional insight into training dynamics and the rationale behind this choice?

**Questions:**

See the Weaknesses part for details.

**Minor Issues:**

Figures 1 and 2 embedded in the PDF fail to render correctly in Safari and some other browsers. The authors are encouraged to check compatibility or provide rasterized alternatives.

---

> ### Author Response · Authors · 2025-11-22
>
> We sincerely thank the reviewer for the thoughtful and detailed comments. We are particularly encouraged by your recognition of our method’s elegance in addressing the visual grounding decay problem, and your assessment that our dataset and results constitute a strong contribution to the field.
>
> We address the following feedback below:
>
> ---
>
> ## **Weakness 1. Dependence on external text-trace generation**
>
> Since there is no canonical ground-truth set of grounding regions for each problem, we manually evaluate the detect-call behavior. Using v1’s reasoning trajectories on the MathVision dataset, we randomly sample 100 trajectories and assess each detect call along four criteria:
>
> - **Appropriateness:** whether the predicted box is relevant to the reasoning step.
> - **Correctness:** whether the box identifies the intended object or region.
> - **Comprehensiveness:** whether the box covers all relevant parts of the target.
> - **Tightness:** whether the box minimizes background and is tightly aligned.
>
> Three annotators independently evaluate each sample for a binary decision. The results show that v1 reliably produces appropriate and correct grounding, while the spatial extent of the boxes is less precise, as reflected in lower comprehensiveness and tightness scores.
>
> | Category          | Maj Vote | Avg Score | Fleiss' $\kappa$ |
> | :---------------- | -------: | --------: | ---------------: |
> | appropriateness   |    90.0% |     87.7% |            0.599 |
> | correctness       |    87.0% |     82.7% |            0.558 |
> | comprehensiveness |    54.0% |     55.7% |            0.689 |
> | tightness         |    40.0% |     49.3% |            0.280 |
>
> ---
>
> ## **Weakness 2. Lack of clarity in visual trace generation**
>
> ### **Algorithmic description**
>
> Due to the space limits, we will only provide a high-level description here. We will supplement this with the code release.
>
> Input: image I, region description T
> Output: bounding box coordinates $b \in R^4$ corresponding to T
>
> 1. **Prepare multimodal input.**
>    Concatenate the image I with a static visual-grounding instruction prompt and feed it to the Qwen2.5-VL model.
>
> 2. **Extract attention with instruction.**
>    From the final decoding position, obtain the cross-attention map A over image tokens.
>    Use a predefined set of layers (selected empirically on a small validation set) and average across heads.
>
> 3. **Extract baseline attention.**
>    Remove object name from the prompt and feed it with the image to the model.
>    Extract the corresponding attention map A′ over image tokens using the same layers and averaging scheme.
>
> 4. **Compute attention contrast.**
>    For each image token, compute the ratio  $R = A / A′,$
>    which serves as a contrastive relevance score over image patches.
>
> 5. **Derive bounding region.**
>    Identify the region with the highest concentration of attention in R.
>    Sweep over multiple candidate crop ratios and, for each ratio, form a bounding region around the peak area.
>    Select the bounding box that yields the **sharpest attention contrast** between its inside and outside regions.
>    Convert this selected region into image-coordinate bounding box b.
>
> Return b.
>
> ### **Statistical analysis**
>
> We provide human evaluation results comparing our attention-based detection method and groundingDINO, a widely used baseline for open-set detection.
>
> We randomly sampled 100 data points from the v1g dataset and 100 bounding boxes from each sampled point. Then, three of the authors conducted a human evaluation process regarding the detection quality. We evaluated them with three aspects: Correctness: the bounding box correctly identifies the referred object/region, Comprehensiveness: The bounding box covers all relevant parts of the object/region, Region Tightness: The bounding box is tight around the object (minimal background).
>
> | Method | Metric | Majority Vote | Avg Score | Fleiss' $\kappa$ |
> |:---|:---|---:|---:|---:|
> | v1g | Correctness | 87.0% | 83.3% | 0.683 |
> | | Comprehensiveness | 56.0% | 55.0% | 0.759 |
> | | Tightness | 46.0% | 44.0% | 0.621 |
> | GroundingDINO | Correctness | 40.0% | 41.0% | 0.711 |
> | | Comprehensiveness | 67.0% | 40.0% | 0.906 |
> | | Tightness | 29% | 32.7% | 0.675 |
>
> As the table above suggests, we argue that the attention-based detection method has distinct detection capabilities compared to GroundingDINO including: (1) geometry (angle 2, line DE), (2) chart & table (bar for ‘grace’), (3) Document & OCR (section: evening prior to the meeting), and (4) knowledge-intensive objects (lateral ventricles in brain ), and (5) referring objects (the second figure)
>
> GroundingDINO has the higher comprehensiveness score because it usually tries to detect the whole image without understanding the given query.

---

> > ### Author Response · Authors · 2025-11-22
> >
> > ## **Weakness 3. Ambiguity in model composition**
> >
> > (L254-259, Sec 4.2) We modified and finetuned a single backbone model (Qwen2.5-VL-7B-Instruct) for v1 to perform textual reasoning and visual grounding in an interleaved manner. A main strength of v1 is that it does not rely on an external module or function for visual grounding. We will revise the description to more thoroughly explain this.
> >
> > ---
> >
> > ## **Weakness 4. Limited interpretability discussion**
> >
> > We acknowledge that the current draft provides only a limited explanation of how v1 uses the pointed visual regions, and we will expand this in the revision. Below is our interpretation of the attention analysis in Figure 5:
> >
> > 1. **Aggregate preference for copied tokens.**
> >    When averaged across layers, attention to the copied tokens is consistently higher than attention to the original image tokens. This shows that once the region is copied into the language context, v1 treats it as a stable and easily accessible reference.
> >
> > 2. **Pre-pointing behavior (Layer 14).**
> >    Attention to the image increases before the point-and-copy operation. This suggests that v1 performs a localization step (identifying where relevant information resides) prior to retrieval.
> >
> > 3. **Post-pointing processing (Layer 2).**
> >    Immediately after the copy operation, the copied tokens display a pronounced fat-tailed attention pattern. This indicates active processing of the retrieved region, with v1 selectively emphasizing specific parts of the copied representation.
> >
> > 4. **High-layer integration (Layer 27).**
> >    At this layer, attention to the copied tokens and the original image tokens becomes similar. While we do not yet have a definitive explanation, one plausible interpretation is that the model performs a consistency check, integrating the retrieved region with the broader image context.
> >
> > In the revised manuscript, we will incorporate these findings into Section 6.2 with expanded explanations, additional qualitative examples that align with the observed attention patterns, and supporting visualizations.
> >
> > ---
> >
> > ## **Weakness 5. Training procedure justification**
> >
> > We source our textual reasoning traces from a previous dataset [1]. We follow the training hyperparameter of the corresponding model (TVC). In practice, multimodal reasoning consists of a much longer token context than typical MLLM training data. This makes the models take longer epochs to fit the reasoning format and patterns. We saw that the model falls into failure modes as repetitions and reasoning without a final answer, using lower epoch values. Note that the point-and-copy behaviour required relatively smaller data for elicitation; typically, we saw the capacity saturated within the first epoch.
> >
> > ---
> >
> > ### **Minor issue**
> > We apologize for the rendering issues with Figures 1 and 2. We are especially grateful that the reviewer took the time to carefully assess our work despite the inconvenience caused by these technical glitches.
> > We have traced down the source of this problem and will fix this in the revision.

---

> > > ### Comment · Reviewer_2iuN · 2025-11-25
> > >
> > > Thank you for your thoughtful response—it has addressed most of my initial questions quite well! As the rebuttal deadline approaches, I wanted to share a friendly suggestion before the final decision. ICLR allows authors to revise the submitted PDF and extend the main text to up to 10 pages during this period. This could be a chance to refine your manuscript further—you might consider integrating the clarifications and details from your rebuttal into the revised PDF to enhance its clarity and completeness.

---

> > > > ### Author Response · Authors · 2025-11-28
> > > >
> > > > We appreciate the reviewer for pointing this out. We have updated the draft accordingly and will incorporate additional revisions, including those raised by other reviewers, before the final draft.
> > > >
> > > > For convenience, the corresponding updates are summarized below:
> > > >
> > > > - **Weakness 1:** Appendix E.2, Table 4
> > > > - **Weakness 2:**
> > > >   - Algorithm: Appendix F
> > > >   - Human study: Appendix E.1, Table 3
> > > > - **Weakness 3:** Section 4.2, L255–261
> > > > - **Weakness 4:** Section 6.2, L470–485
> > > >   - Additional qualitative samples are being prepared
> > > > - **Weakness 5:** Appendix A, L781–786

---

### Official Review · Reviewer_zMQY · 2025-11-01

**Soundness:** 3
**Presentation:** 2
**Contribution:** 2
**Rating:** 6
**Confidence:** 3

**Summary:**

This paper addresses a critical issue in vision-language models (VLMs): the diminishing influence of visual tokens as the chain-of-thought (CoT) lengthens. To mitigate this, the authors introduce a novel training dataset, v1g, and a resulting model, v1. The v1g dataset is constructed from VQA samples, where CoT sequences are augmented by interleaving relevant visual tokens, copied directly from the input, to reinforce visual grounding. For instance, a reasoning step like "query z" is explicitly followed by the visual tokens corresponding to object "z". The authors fine-tune a base VLM on V1G to obtain the V1 model. Experimental results on benchmarks, including MathVista, MathVision, and MathVerse, demonstrate the effectiveness of the proposed approach.

**Strengths:**

1. This paper proposes a novel training dataset named v1g to address a critical issue in VLMs: the diminishing influence of visual tokens as the chain-of-thought (CoT) length increases.

2. By fine-tuning VLMs on v1g, the authors obtain the v1 model. Experimental results on three mathematical VQA benchmarks demonstrate the effectiveness of the proposed method.

**Weaknesses:**

1. The evaluation of the proposed method is currently limited to mathematical VQA tasks. Its performance on other domains remains unclear, and further experiments on general VQA benchmarks are needed to assess the generalization capability of both the dataset and the fine-tuned VLM.

2. Does the reuse of visual tokens in the chain-of-thought introduce additional computational overhead compared to text-only CoT? A quantitative comparison of computational costs between the proposed method and the baseline would help clarify this practical concern.

**Questions:**

1. How does the method generalize to other VQA domains?

2. What is the computational overhead compared to baseline methods?

---

> ### Author Response · Authors · 2025-11-22
>
> We sincerely appreciate the reviewer’s insightful comments. We are encouraged by your recognition of the novelty of the v1g dataset in addressing visual grounding decay and the substantiated effectiveness of our method. We address the concerns below:
>
> ---
>
> ## **Weakness 1. Focus on mathematical reasoning**
>
> We recognize the value of broader, general-domain evaluations. However, given the substantial cost of constructing multi-purpose multimodal models, our study focuses on the standard domain used to assess multimodal reasoning.
>
> 1. **Mathematics is a canonical testbed for multimodal reasoning.**
>
> Leading benchmarks in this area [1-4] predominantly evaluate mathematical and geometric problems because they provide well-defined structures, unambiguous ground truth, and controlled settings for analyzing reasoning behaviours. As mathematics is also a foundational domain for textual reasoning [5], focusing on this domain does not limit the value of our experiments; rather, it aligns with how multimodal reasoning is conventionally evaluated.
>
> 2. **Multi-task model building is costly both in terms of data and computation.**
>
> Our dataset is built on top of text-only reasoning traces [6], which primarily target verifiable domains such as mathematics. To extend our approach to general-purpose settings, we would first need large-scale, high-quality reasoning traces across diverse domains; resources that are currently unavailable at the necessary scale. In addition, training a unified multi-domain model is computationally demanding.
>
> Given these constraints, we prioritize demonstrating the effectiveness of grounded reasoning and pointing-based semantic grounding within the widely adopted evaluation protocol.
>
> ---
>
> ## **Weakness 2. Computational overhead from visual grounding**
>
> The additional computational cost is not extensive.
> 1. **Architectural overhead.**
>
>
> v1 introduces only a single linear head applied to the input image tokens (256 tokens for a 448×448 image processed by the Qwen2VL processor). This is negligible compared with the textual vocabulary size (131,072-dimensional in case of Qwen2VL).
>
>
> 2. **Sequence-length overhead.**
>
>
> Our SFT trajectories suppress revisitation when the same object was visited recently (within the last 400 tokens). Consequently, in both training data and model outputs, copied visual tokens rarely exceed 60\% of the textual token count, keeping the total reasoning length within approximately 1.6 times of the corresponding text-only traces.
>
> ---
>
> ## **Question 1. Generalization to other domains**
>
> Our method is readily extendable to other domains. The primary bottleneck is data availability: grounded reasoning trajectories are difficult to obtain at scale. Given access to such data, the extension is straightforward: training the same framework would suffice.
>
> ---
>
> ### **References**
>
> [1] OpenCompass Leaderboard (https://huggingface.co/spaces/opencompass/Open_LMM_Reasoning_Leaderboard)
>
> [2] MathVerse, ECCV 2024 (https://arxiv.org/pdf/2403.14624)
>
> [3] MathVision, NeurIPS D&B 2024 (https://arxiv.org/pdf/2402.14804)
>
> [4] MathVista, ICLR 2024 (https://arxiv.org/pdf/2310.02255)
>
> [5] AIME 2025 (https://huggingface.co/datasets/MathArena/aime_2025)
>
> [6] TVC, ACL 2025 (https://arxiv.org/pdf/2503.13360)

---

### Official Review · Reviewer_7bZK · 2025-11-03

**Soundness:** 4
**Presentation:** 3
**Contribution:** 3
**Rating:** 6
**Confidence:** 5

**Summary:**

MLLMs often lose visual grounding as reasoning unfolds, since they process images only once before generating purely textual inferences. This paper proposes v1, a lightweight extension that enables active visual referencing through a simple point-and-copy mechanism. v1 allows the model to dynamically select relevant image patches and copy their embeddings into the reasoning stream, keeping inference grounded in perceptual evidence. To train this ability, the authors construct v1g, a dataset of 300 K multimodal reasoning traces with interleaved visual-grounding annotations. Evaluated on three multimodal mathematical reasoning benchmarks, v1 consistently surpasses comparable baselines, particularly on tasks requiring fine-grained visual grounding. These results demonstrate that dynamic visual access via point-and-copy offers an effective and efficient mechanism for grounded multimodal reasoning.

**Strengths:**

1)	The paper identifies a concrete weakness in current MLLMs, the visual grounding decay problem during reasoning, and provides empirical evidence to substantiate this observation.
2)	The proposed point-and-copy approach offers a simple yet elegant solution to dynamically re-access visual representations during reasoning, conceptually bridging textual reasoning and visual perception.
3)	The method introduces minimal additional parameters (two linear heads), making it easy to integrate into existing MLLMs without significant computational or architectural overhead.

**Weaknesses:**

1)	Empirical validation is confined to MathVista/MathVision/MathVerse. Broader domains (charts beyond math, documents, VQA, OCR-heavy tasks) are not evaluated, limiting claims of generality. Consider adding non-math benchmarks.
2)	Although the method is said to be generally compatible, experiments are instantiated only on Qwen2.5-VL-7B; cross-backbone results (e.g., InternVL/LLaVA variants) would strengthen the case for portability.
3)	The paper offers only brief training descriptions, making it difficult to assess reproducibility and generalization. More details on optimization settings, schedule, and potential use of reinforcement-style methods (e.g., R1-type reasoning training) would help clarify robustness and strengthen confidence in the reported results.
4)	In Figure 3(b), although the ratio of attention to salient regions decreases as generation progresses, the overall attention level remains relatively high. It is unclear how the authors interpret this result: does a high absolute ratio still indicate grounding decay, or could it reflect stable focus on visual areas?
5)	In Figure 3(a), the decrease of attention on visual tokens with longer generation steps seems natural for autoregressive reasoning, where attention gradually shifts from perception to internal memory. The key question is not whether decay occurs, but how fast it happens. The paper interprets the observed decline as evidence of model deficiency, yet does not justify why the speed of attention decay indicates unreasonable behavior. A more principled analysis or comparison across models with different decay rates would make this argument more convincing.
6)	Swapping Figures 2 and 3 would improve narrative clarity: the paper should first show the attention-decay problem (Fig. 3) before introducing the proposed point-and-copy solution (Fig. 2).

**Questions:**

Please refer to the weaknesses.

---

> ### Author Response · Authors · 2025-11-22
>
> We sincerely thank the reviewer for the constructive and positive assessment. We address the feedback below:
>
> ---
>
> ## **Weakness 1. Focus on mathematical reasoning**
>
> We recognize the value of broader, general-domain evaluations. However, we focus on the standard domain used to assess multimodal reasoning.
>
> 1. Mathematics is a canonical testbed for multimodal reasoning.
>
> Leading benchmarks in this area [1-4] predominantly evaluate mathematical and geometric problems because they provide well-defined structures, unambiguous ground truth, and controlled settings for analyzing reasoning behaviours. As mathematics is also a foundational domain for textual reasoning [5], focusing on this domain does not limit the value of our experiments; rather, it aligns with how multimodal reasoning is conventionally evaluated.
>
> 2. Multi-task model building is costly both in terms of data and computation.
>
> Our dataset is built on top of reasoning traces, which primarily target verifiable domains such as mathematics [6]. To extend our approach to general-purpose settings, we would first need large-scale, high-quality reasoning traces across diverse domains; resources that are currently unavailable at the necessary scale. Additionally, training a unified, multi-domain model is computationally demanding.
>
>
> Given these constraints, we prioritize demonstrating the effectiveness of grounded reasoning and pointing-based semantic grounding within the widely adopted evaluation protocol.
>
> ---
>
> ## **Weakness 2. Cross-backbone experiment**
>
>
> We agree that cross-backbone validation is important. To address this, we are running additional experiments on LLaVA-OneVision-1.5-4B. On early checkpoints, the v1-adaptation improves MathVision accuracy from 23.4 to 31.3 (+7.9%). We will continue training the baselines and include a complete ablation study in the final draft.
>
> ---
>
> ## **Weakness 3. More detail on the training process**
> Our setup uses an effective batch size of 64, AdamW (lr 3e-5), linear decay, and a 0.03 warmup ratio. All experiments use bf16 and DeepSpeed. We will expand Appendix A to incorporate the details and release code for full reproducibility.
> Regarding RL, our framework places no restriction on applying post-training reinforcement methods to v1, since v1 uses no external modules during generation. However, R1-style long-reasoning training is computationally intensive, and we view it as a direction for future work.
>
> ---
>
> ## **Weakness 4. Interpretation of Figure 3(b)**
>
> We apologize for our lack of explanation. Figure 3(b) reports a *relative* attention measure, defined as $R = \frac{\frac{1}{|S|}\sum_{i \in S} A_i}{\frac{1}{|I|}\sum_{j \in I} A_j}$, where $A$ denotes attention scores, $S =$ set of salient-region patches and $I =$ set of all image patches.
>
> Therefore, a value of 0.8 does **not** indicate strong grounding. It means that a patch inside the salient region receives **less** attention than an average patch in the image. Hence, the model distributes attention more heavily toward non-salient regions rather than the evidence-relevant area.
>
> Prior work [7] also reports that sharper, more concentrated attention patterns correlate with higher model accuracy.
>
> ---
>
> ## **Weakness 5. Interpretation of Figure 3(a)**
>
> We agree that attention decay is not inherently harmful; the problem is premature decay before sufficient visual information is extracted. We provide two supporting pieces of evidence:
>
> 1. Causal evidence.
>
> A prior study [6] demonstrated that simply re-inserting the input image during reasoning improves downstream performance, indicating that visual information is *not* fully extracted early in the chain. Table 2 from that work reports:
>
> | Method          | MathVista | MathVision | MathVerse |
> | :---------------- | -------: | --------: | --------: |
> | Baseline       | 63.5 | 19.8 | 31.6 |
> | Visual revisit | 66.2 | 22.3 | 34.7 |
>
> 2. Qualitative evidence.
>
> We manually inspected 100 random trajectories generated by TVC on MathVision. Only 29  outputs correctly captured all relevant visual details within the first 500 tokens. Machine-based classification (Gemini-2.5-Flash) produced an identical rate. These results show essential visual details are not fully captured before attention decay happens.
>
> We plan to expand this study across more long reasoning models in the final draft.
>
> ---
>
> ## **Weakness 6. Figure positioning**
>
> We appreciate the reviewer’s suggestion and promise to update this on the final revision.
>
> ---
>
> ### **References**
>
> [1] OpenCompass Leaderboard (https://huggingface.co/spaces/opencompass/Open_LMM_Reasoning_Leaderboard)
>
> [2] MathVerse, ECCV 2024 (https://arxiv.org/pdf/2403.14624)
>
> [3] MathVision, NeurIPS D&B 2024 (https://arxiv.org/pdf/2402.14804)
>
> [4] MathVista, ICLR 2024 (https://arxiv.org/pdf/2310.02255)
>
> [5] AIME 2025 (https://huggingface.co/datasets/MathArena/aime_2025)
>
> [6] TVC, ACL 2025 (https://arxiv.org/pdf/2503.13360)
>
> [7] ADAPTVIS, ICML 2025 (https://arxiv.org/pdf/2503.01773)

---

### Author Response · Authors · 2025-12-02
**Summary of Review Process and Outstanding Issues**

To facilitate assessment of the review process, we summarize the overall reviewer landscape and our responses to the remaining concerns.


---


## **Main contribution of the paper**


- Identifies visual grounding decay as a critical failure mode in Multimodal Large Language Models (MLLMs), where attention to relevant visual regions diminishes as reasoning chains lengthen.
- Introduces v1, a model equipped with a point-and-copy mechanism that dynamically identifies relevant image patches and copies their embeddings back into the reasoning stream without external modules.
- Constructs v1g, a dataset of 300K multimodal reasoning traces with interleaved visual grounding annotations.
- Demonstrates that v1 outperforms baselines on mathematical reasoning benchmarks (MathVista, MathVision, MathVerse) and provides controlled analyses of attention patterns.


---


## **Overall reviewer stance**



**Three of four reviewers (7bZK, zMQY, 2iuN) expressed positive evaluations and supported acceptance.** These reviewers highlighted the elegance of the point-and-copy mechanism, the value of the **v1g dataset**, and the soundness of the motivation regarding grounding decay. Reviewer 2iuN was particularly engaged, noting that the clarifications regarding human validation and algorithmic descriptions addressed their initial concerns, leading to a suggestion to update the manuscript during the rebuttal period.


---

## **Primary remaining concern**


The concerns stem from differing views on what constitutes the paper’s primary contribution. Reviewer db5f emphasizes architectural novelty and interprets the 2.6-point improvement over the ablation as insufficient to justify the pointing mechanism.


In contrast, we and the other three reviewers evaluate the work based on its integrated contributions: the v1g dataset, the grounded-reasoning framework, and the approach to mitigating grounding decay. From this standpoint, the ablation’s strong performance highlights the value of the grounded-reasoning framework itself, which is the central contribution of the paper.


We also note that the 2.6-point improvement is not trivial: it corresponds to an approximately 11% relative gain over the backbone’s 23.6 accuracy. Moreover, this additional gain enables v1 to reach performance comparable to the 72B-scale reasoner (QvQ-72B) from which the SFT trajectories were originally derived, underscoring the practical significance of the pointing mechanism.


---


## **Resolution of other concerns**

We successfully addressed several technical questions raised by the positive reviewers:
- **Generalization (Reviewer 7bZK, zMQY)**: We clarified that while our experiments focus on math (a standard testbed for reasoning), the method is domain-agnostic. We provided new preliminary results on LLaVA-OneVision showing a +7.9% gain on MathVision, demonstrating cross-backbone transferability.
- **Visual Trace Quality (Reviewer 2iuN, db5f)**: We conducted a human evaluation study (N=100) comparing our attention-based bounding boxes against GroundingDINO. Our method achieved significantly higher correctness (87% vs 40%) on complex reasoning targets (e.g., specific geometric angles, charts), validating the quality of our automatic annotation pipeline.
- **Visual Attention Decay (Reviewer 7bZK, db5f)**: We clarified the metric in Figure 3(b) (relative attention ratio) and cited prior work (AdaptVis) and our own qualitative analysis (only 29/100 correct visual extractions in early layers) to prove that premature attention decay is a genuine bottleneck, not just a feature of autoregressive models.


---




We are happy to provide additional details if they would assist the AC’s assessment.

---

### Meta-Review · Area_Chair_2V6L · 2026-01-06

**Summary:**

Overall, this work received borderline-but-leaning-positive reviews initially, with likely a slightly improved score after rebuttal but some reviewers likely unconvinced. It's generally an interesting idea (though the human inspiration is a little bit tenous; the mechanisms in humans are clearly not point-and-copy) with decent evaluations and results.

The main remaining concerns (though not major) that I consider not fully addressed/resolved/convincing, based on reviewer responses to rebuttals, plus my own assessment, are regarding limitations in the scope/generality (e.g. focus on math-related VQA) and questions about tool use.

While these issues are likely addressable in a subsequent revision/submission, for the work in its current form, given the highly-selective nature of ICLR, I'm afraid I do not recommend acceptance at present, particular in comparison to other submissions.

**Reviewer Concerns:**

Pls see above.

**Reviewer Scores:**

-- 2iuN: potentially raised score to 8 (was happy with rebuttal but did not explicit mention raising score).
-- db5f: likely maintain at 4 (was not convinced by first round of rebuttal).
-- zMQY and 7bZK: I hazard they would maintain their positive-leaning 6s.

---

### Decision · Program_Chairs · 2026-01-26

Reject